# SPEECH ROBUST BENCH: A ROBUSTNESS BENCHMARK FOR SPEECH RECOGNITION

**Muhammad A. Shah**[*1] **David Solans**[2] **Mikko A. Heikkilä**[2,3] **Bhiksha Raj**[1] **Nicolas Kourtellis**[2]

## ABSTRACT

As Automatic Speech Recognition (ASR) models become ever more pervasive, it is important to ensure that they make reliable predictions under corruptions present in the physical and digital world. We propose `Speech Robust Bench` (SRB), a comprehensive benchmark for evaluating the robustness of ASR models to diverse corruptions. SRB is composed of 114 challenging speech recognition scenarios which largely cover the range of corruptions that ASR models may encounter when deployed in the wild. We use SRB to evaluate the robustness of several state-of-the-art ASR models and observe that model size and certain modeling choices such as the use of discrete representations, or self-training appear to be conducive to robustness. We extend this analysis to measure the robustness of ASR models on data from various demographic subgroups, namely English and Spanish speakers, and males and females. Our results revealed noticeable disparities in the model's robustness across subgroups. We believe that SRB will significantly facilitate future research towards robust ASR models, by making it easier to conduct comprehensive and comparable robustness evaluations.

## 1 INTRODUCTION

As novel ML models continue to be developed *and deployed* at an ever-increasing rate, it has become crucial to ensure their robustness to challenging real-world scenarios, where corruptions arising from a myriad of sources, including the environment, sensing apparatus, and even malicious actors are present. To this end, prior works have developed comprehensive robustness benchmarks, particularly for vision (Hendrycks & Dietterich, 2019; Hendrycks et al., 2021a;b; Croce et al., 2020) and natural language processing models (Wang et al., 2021a; 2022b), that evaluate a model's performance under a variety of challenging scenarios. These benchmarks have proven to be invaluable to the advancement of research into more robust models because (1) they unify robustness evaluations, thus enabling meaningful comparisons across models and allowing progress to be accurately tracked, and (2) they make it easier for researchers to comprehensively evaluate the robustness of their models by aggregating a diverse and representative set of scenarios, and methods of simulating them, in a single benchmark.

While several robustness benchmark datasets exist for Automatic Speech Recognition (ASR) models (Barker et al., 2017; Kraaij et al., 2005; Wichern et al., 2019; Reddy et al., 2020; Cosentino et al., 2020; Hershey et al., 2016; Chen et al., 2020; Snyder et al., 2015; Kinoshita et al., 2013; Ko et al., 2017; Nakamura et al., 2000; Jeub et al., 2009), none of the currently existing ones are in any sense comprehensive, because each benchmark measures the model's robustness w.r.t. to one or a few specific types of corruptions or scenarios, which puts the onus on model developers to find and collect all the relevant benchmarks to evaluate their model comprehensively. This has often resulted in model developers evaluating their models on disparate benchmarks (Radford et al., 2023; Wen et al., 2016; Chen et al., 2022; Likhomanenko et al., 2020), which makes it hard to reliably compare performance and robustness across models. Recently, Huggingface Open ASR Leaderboard (Srivastav et al., 2023) has sought to unify ASR model evaluations by developing a benchmark consisting of several real-world speech datasets. Although evaluating models on exclusively natural data may accurately reflect average case real-world performance, it is generally not informative about the specific types of corruptions the models are weak against, because the noise sources present in these datasets are not

---

*email: mshah1@cmu.edu [1] Carnegie Mellon University, [2] Telefonica Research, [3] University of Helsinki; M. Shah did this work during an internship at Telefonica Research; N. Kourtellis is now at Keysight Technologies.

controlled or even fully known. For example, the crowdsourced recordings in Common Voice (Ardila et al., 2019) contain a variety of distortions including sensor noise from low-quality equipment, background noise, and mispronunciation by non-native speakers. Furthermore, digital perturbations like special effects, computer-generated speech, and adversarial examples, that may be prevalent in digital content are largely overlooked by existing benchmarks.

In this paper, we propose `Speech Robust Bench` (SRB), a benchmark for comprehensively evaluating the robustness of ASR models to input perturbations and corruptions. SRB is designed to address the aforementioned major shortcomings of existing ASR robustness benchmarks, i.e., that (1) they are often specialized and thus are not individually comprehensive, (2) even taken together, they overlook important challenging scenarios, like special effects and adversarial attacks, and (3) may not reveal the specific weaknesses of the models. SRB addresses these shortcomings by evaluating ASR models under a comprehensive set of challenging scenarios, using recordings that are either are recorded under specific scenarios, and thus are inherently "noisy", or recordings that are digitally perturbed to simulate the various scenarios. SRB uses real recordings of accented speech and inter-personal conversations to evaluate robustness to articulatory and lexical variability. We take care to ensure that the recordings are clean and do not have any other corruption that may confound the results. To digitally simulate challenging scenarios, we curate a large comprehensive bank of 114 perturbations that represent common distortions arising from the environment, recording equipment, special effects, computer-generated speech, and adversarial attacks that are often overlooked by existing benchmarks.

To highlight the need for and the benefits of doing systematic and comprehensive robustness assessment, we evaluate the robustness of several popular ASR models (§4.1) using SRB. We observe that Whisper (Radford et al., 2023) is the most robust *on average* among the models we tested, even outperforming the more recent Canary (NVIDIA). We conduct further analyses to disentangle the effects of model and training data size, revealing that larger models tend to be more robust than smaller models, even if the latter are trained on significantly more data. We further extend our analysis by evaluating the models' robustness for the various population sub-groups, namely, English and non-English (Spanish) speakers, and male and female speakers. We find that significant disparities exist across these sub-groups, thus identifying issues with fairness of the models and highlighting areas where future work could provide clear improvements. Besides pinpointing robustness issues, this demonstrates the utility of SRB for fairness evaluations that consider robustness disparities across models as well.

To summarize we make the following contributions:

- We present SRB, a comprehensive robustness benchmark for ASR models, enabling direct and easy comparison of robustness evaluations between models to facilitate progress.
- We demonstrate the use of SRB by conducting a fine-grained robustness analysis for several popular models. We extend our analysis by using SRB to uncover disparities in the robustness of ASR for various sub-groups of speakers. This highlights the broad utility of this benchmark to the field of trustworthy AI.
- To facilitate out-of-the-box robustness evaluations for the community, we have publicly released a large dataset [1] containing perturbed versions of LibriSpeech (Panayotov et al., 2015) test-clean, Spanish, and French and German test sets of Multilingual LibriSpeech (Pratap et al., 2020), as well as accented speech from common voice, and segemented near- and far-field audios from CHiME-6 (Reddy et al., 2020) and AMI(Kraaij et al., 2005).
- We release our code with clear documentation to enable reproducibility and extensibility. [2]

## 2 RELATED WORK

### 2.1 ROBUST AUTOMATIC SPEECH RECOGNITION

Several techniques have been proposed for making Automatic Speech Recognition (ASR) models robust to input perturbations, such as noise and other signal corruptions (Li et al., 2014). We can divide these techniques into two high-level categories: i) model-based and ii) feature-based. Model-

---

[1]data: `https://huggingface.co/datasets/mshah1/speech_robust_bench_public`
[2]code:`https://github.com/ahmedshah1494/speech_robust_bench`

based techniques modify the models to make them more robust. Examples of such approaches include adapting pre-trained models (Yu et al., 2009; Juang & Rahim, 1996), de-noising the audio before processing (Mohammadiha et al., 2013; Wilson et al., 2008), and training ASR models on noisy data (Likhomanenko et al., 2020). Since model-based strategies generally require access to noisy data (Li et al., 2014), they are most effective if the sources of noise, and/or the exact environment in which the ASR model will be deployed in are known, and one can gather data to represent those. Feature-based approaches, on the other hand, involve developing handcrafted features, invariant to noise and corruptions in the signal (Li et al., 2014). Several of these features are inspired by biological audition (Kim & Stern, 2016; Hermansky et al., 1991; Hermansky & Sharma, 1998), while others use signal processing techniques (Li et al., 2014). Generally, these methods are designed to extract the components of the audio signal salient for speech production and perception, while discarding irrelevant components (Stern & Morgan, 2012). Consequently, they do not require precise knowledge of the environment and noise distributions. Recently, however, handcrafted features have fallen out of favor, and have been replaced by features learned via end-to-end training of deep learning models on large amounts of data (Baevski et al., 2020; Hsu et al., 2021a; Likhomanenko et al., 2020; Radford et al., 2023). Proponents of these techniques posit that models trained on larger datasets become more robust. Our evaluations in § 4 reveal that there are several input perturbations against which smaller models trained on less data outperform larger models trained on more data.

## 2.2 ADVERSARIAL ROBUSTNESS

Adversarial perturbations can change the response of a model when added to their inputs, but are either imperceptible to humans or perceptually and semantically irrelevant enough to be ignored by them (Szegedy et al., 2014; Goodfellow et al., 2014). Adversarially perturbed inputs are known as *adversarial attacks*. They can be targeted (aiming to change a prediction to a specific incorrect class), or un-targeted (aiming to change a prediction to any incorrect class (Akhtar et al., 2021)). The design of adversarial attacks is determined by the level of knowledge the attacker is assumed to have about the target model. Attacks that assume full knowledge of the target model's architecture and weights (*white-box* threat model) often use gradient-based optimization techniques (Szegedy et al., 2014; Goodfellow et al., 2014; Madry et al., 2018; Laidlaw et al., 2021; Akhtar et al., 2021). Attackers who do not have any knowledge of the target model's architecture and only have query access to it (*black-box* threat model) typically use gradient-free optimization methods (Wang et al., 2022a; Andriushchenko et al., 2020; Wicker et al., 2018; Chen et al., 2017; Zhao et al., 2020; Vo et al., 2022). An intriguing property of adversarial perturbations is that they transfer between models (Papernot et al., 2016), and inputs (Akhtar et al., 2021; Neekhara et al., 2019), i.e. perturbations designed for one model/input may be effective against others as well. Our SRB includes two types of white box adversarial attacks; those that generate perturbations: 1) specific to each input (Madry et al., 2018), 2) that cause models to mis-transcribe multiple inputs (Neekhara et al., 2019).

## 2.3 ROBUSTNESS BENCHMARKS FOR SPEECH

While several robustness benchmark datasets exist for ASR models, they, unfortunately, suffer from three major shortcomings that make it difficult to perform robustness evaluations in a way that is comprehensive, sufficiently fine-grained, and comparable across models.

Firstly, we find that *existing benchmarks do not comprehensively evaluate robustness*. While past works such as (Pearce & Picone, 2002) did indeed propose benchmarks containing diverse perturbations, they were limited to relatively simple data unsuitable for modern ASR models. However, many recently proposed benchmarks measure the robustness in one or a few specific types of scenarios. For example, some datasets (Kinoshita et al., 2013; Nakamura et al., 2000; Jeub et al., 2009; Ko et al., 2017) focus exclusively on reverberant speech, while others focus only on multi-speaker scenarios (Kraaij et al., 2005; Barker et al., 2017), environmental noise (Snyder et al., 2015; Reddy et al., 2020; Wichern et al., 2019; Hirsch & Pearce, 2000), or accented speech (Lander, 2022; Shi et al., 2021). Furthermore, certain types of digital perturbations, like special effects, computer-generated speech, and adversarial examples, which humans are usually invariant to (and thus ASR models are expected to be as well) are largely overlooked by existing benchmarks. *SRB evaluates robustness under a wide range of scenarios, including those represented in prior works, and novel ones that have not yet received due attention, thus enabling comprehensive robustness evaluations via a single benchmark.*

Secondly, since there are several robustness benchmarks that researchers can choose from, the onus is on model developers to collect all relevant benchmarks and comprehensively evaluate their model. This has often resulted in model developers evaluating their models on disparate benchmarks (Radford et al., 2023; Wen et al., 2016; Chen et al., 2022; Likhomanenko et al., 2020), which makes it hard to reliably compare performance and robustness across models. For example, several works have used subsets of existing benchmarks to evaluate the model robustness (Radford et al., 2023; Wen et al., 2016; Chen et al., 2022), or have evaluated the robustness of models by computing their transcription accuracy on multiple natural speech datasets (Likhomanenko et al., 2020; Radford et al., 2023; Hsu et al., 2021b). *SRB covers a wide range of challenging ASR scenarios in a single benchmark, it obviates the need to select from various benchmarks, and, thus, can unify robustness evaluations across studies and make them comparable.*

Finally, robustness benchmarks that try to mitigate the above two shortcomings are often too coarse to reveal the specific scenarios and corruptions the model(s) struggle against. For example, Huggingface Open ASR Leaderboard(OAL, Srivastav et al. 2023) evaluates models on several real-world speech datasets, which may accurately reflect their *average-case* real-world performance, it may not inform about the specific type of perturbations the models are weak against. This is because noise sources present in these datasets are not controlled or even fully known. For example, the crowdsourced recordings in Common Voice (Ardila et al., 2019) contain a variety of distortions including sensor noise from low-quality equipment, background noise, and mispronunciation by non-native speakers. If a model struggles with Common Voice, identifying the specific types of noise it is sensitive to is not a straightforward task. *In the design of SRB, we have ensured that the sources of noise (and variance, in general) in each utterance are limited and known, which allows users to pinpoint specific scenarios and/or perturbations under which their models struggle (e.g. Fig. 3 and § 4).*

## 3 SPEECH ROBUST BENCH

`Speech Robust Bench` (SRB) evaluates the robustness of ASR models by a three-step process consisting of (1) scenario simulation, (2) transcription, and (3) metrics computation, as shown in Fig. 1: first, various challenging speech recognition scenarios are simulated by applying a large bank of synthetic perturbations to clean speech datasets (§ 3.1), as well as by using inherently noisy speech datasets with limited and known sources of real noise and variations that

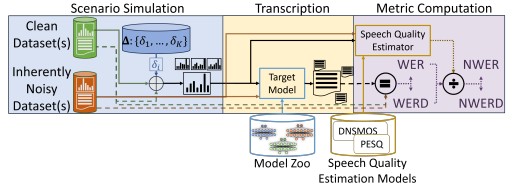

Figure 1: An illustration of the processes involved in using our benchmark to evaluate the robustness of ASR models.

are difficult to simulate. Next, the perturbed recordings, the original clean recordings, and the recordings with inherent noise are transcribed using the target ASR model. Finally, the predicted and reference transcripts are compared, and the accuracy and robustness of each model in each setting is captured with various metrics (§ 3.2). To account for the differences in the level of difficulty between scenarios, we also estimate speech quality scores using appropriate models and use them to calculate normalized metrics.

### 3.1 SCENARIO SIMULATION

The various speech recognition scenarios simulated by SRB are taxonomized in Fig. 2, and can be divided into six high-level categories, namely (1) clean speech, (2) social gatherings, (3) speech variations, (4) environmental effects, (5) digital augmentations, and (6) adversarial attacks. The scenarios are described briefly below, while more details are given in Appendix A.

**(1) Clean speech**: SRB uses clean speech for two purposes: to benchmark the baseline accuracy of ASR models, and to simulate various challenging scenarios by perturbing it. Clean

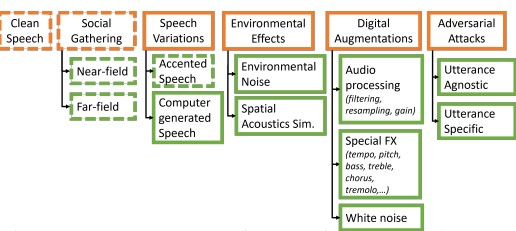

Figure 2: Taxonomy of scenarios currently represented in SRB. Scenarios in dashed boxes have real-world recordings, while scenarios in solid boxes are simulated by digitally adding perturbations.

speech is drawn from Librispeech (Panayotov et al., 2015) test-clean, TEDLIUM (Hernandez et al., 2018) release 3 test and MultiLingual LibriSpeech (MLS) (Pratap et al., 2020) test. LibriSpeech contains professional recordings of English audio books. Meanwhile, TEDLIUM contains professional recordings of English TED talks and provides lexical and phonetic diversity which LibriSpeech may lack. To increase the applicability of SRB to non-English and multi-lingual models we also include Spanish speech from MLS, which contains professionally recorded audio books in several languages.

**(2) Social Gatherings**: The ability to transcribe speech from semi-formal or informal settings, as well as far-field audio is useful for models used in meeting rooms, smart homes, and even subtitle generation, thus SRB includes English speech from dinner parties and meetings recorded by (2.1) *near-* and (2.2) *far-field* mics from CHiME-6 (Barker et al., 2017) and AMI (Kraaij et al., 2005).

**(3) Speech Variations:** ASR models must remain accurate under variations in pronunciation and prosody to serve diverse speakers. We therefore include (3.1) *clean accented speech*[3] from English and Spanish subsets of Common Voice 17 (CV17, Ardila et al. 2019) in SRB. To provide additional prosodic variability and to represent the increasing pervasiveness of generative AI, we also include synthetic speech generated by YourTTS (Casanova et al., 2022) (English) and Bark Suno (b) (Spanish) from transcripts of the three clean datasets from scenario (1) in the voices of English and Spanish speakers from VCTK (Yamagishi et al., 2019) and Bark Speaker Library (v2, Suno a), respectively.

The following scenarios involve synthetic perturbation of all three clean datasets from scenario (1).

**(4) Environmental Effects:** While noisy real speech datasets like CommonVoice (Ardila et al., 2019) and Switchboard (Godfrey et al., 1992) exist, the noise in them is not controlled or even known. Thus in SRB, we perturb clean speech to simulate (4.1) *environmental noise*, and (4.2) *spatial acoustics*. Concretely, we add *real* environmental noise from ESC-50 (Piczak, 2015), MS-SNSD (Reddy et al., 2019), MUSAN (Snyder et al., 2015) and WHAM! (Wichern et al., 2019) at Signal-to-Noise Ratios (SNR) of 10, 20, 30 and 40 dB. To simulate spatial acoustics, we add echo via SoX[4] and simulate Room Impulse Response (RIR) via convolution with real and simulated RIRs from Ko et al. (2017).

**(5) Digital Augmentations:** Digital media often undergoes processing and contains special effects, which are therefore included in SRB. Specifically, we include *standard audio processing operations* like amplitude gain, resampling, lowpass, and highpass filtering, (2.2) *special effects* like bass gain, treble gain, tempo increase, tempo decrease, speed increase, speed decrease, pitch increase, pitch decrease, chorus, tremolo, and phaser, and (2.3) *Gaussian white noise*.

**(6) Adversarial Attacks:** Models used in high-stakes settings are prime targets for adversaries and thus must resist attempts to compromise their accuracy. We use two types of adversarial attacks in SRB: (2.1) *utterance-specific* and (2.2) *utterance-agnostic* attacks. The utterance-specific attack searches for a perturbation $\delta$, for a given speech recording $x$, such that a given model maximally mistranscribes it. To find $\delta$, we follow Madry et al. (2018) and use projected gradient descent to solve $\max_{\delta:\text{SNR}(\delta,x)\leq\epsilon} \mathcal{L}(M(x), y^*)$, where $\mathcal{L}$ is a differentiable loss function, like CTC-Loss, between the model's output $M(x)$ and the true transcript $y^*$, with $\epsilon \in [10, 40]$. The utterance-agnostic attack is similar to the utterance-specific attack, except $\delta$ is optimized over a held-out *set*, $\mathcal{X}^{\text{dev}}$, instead of each test utterance. This represents a more realistic scenario where an attacker tries to mount a denial-of-service attack against an ASR model by introducing utterance-agnostic perturbation at some point in the transcription pipeline. We use the method of Neekhara et al. (2019) to find $\delta : \mathbb{E}_{x \in \mathcal{X}^{\text{dev}}}\text{SNR}(\delta, x) \leq \epsilon \in [10, 40]$ such that $CER(\{x + \delta | x \in \mathcal{X}^{\text{dev}}\}) > \tau$ (see Alg. 1), where $\mathcal{X}^{\text{dev}}$ is the dev split of LibriSpeech, TEDLIUM and MLS.

**Note of Extensibility and Usage:** We have released our source code with instructions for reconstructing the data in SRB, and reproducing the results of this paper ( § 1). SRB can easily be extended to other languages and speech datasets using the provided scripts for extracting accented speech from any language in CV17, and for simulating scenarios 3.2-6 on any speech recording or dataset.

We have also publicly released the data for all the above scenarios, except adversarial attacks and perturbed TEDLIUM recordings, on Huggingface Hub (see footnote 1). We made these exceptions because TEDLIUM's license prohibits the distribution of derivatives, and the adversarial attacks must be computed separately for each target ASR model. *To the extent possible, we encourage users to evaluate their models on the publicly released data to ensure reproducibility.*

---

[3]Accent annotations (excluding US English), and a DNSMOS score $\geq 3.4$ (Reddy et al., 2020).

[4]Available from `https://sourceforge.net/projects/sox/`.

## 3.2 METRICS

SRB measures the *utility* of the model with the widely used **Word Error Rate**. WER is computed as the word-level edit distance between the reference and the predicted transcripts, normalized by the length of the reference (see Appendix B for formal definitions). To measure the *robustness* of the model under challenging scenarios, we use **WER Degradation** (WERD), computed as $WER(\mathcal{X}_s) - WER(\mathcal{X})$, where $\mathcal{X}$ and $\mathcal{X}_s$ are datasets containing clean speech and speech from scenario $s$, respectively. For scenarios (1)-(3.1) (see § 3.1), $\mathcal{X}_s$ is an inherently noisy dataset, and $\mathcal{X}$ will be LibriSpeech for English and Multi-Lingual LibriSpeech for Spanish. For scenarios (3.2)-(6), $\mathcal{X}$ is a clean dataset, and $\mathcal{X}_s$ is a perturbed version of $\mathcal{X}$.

When aggregating metrics (WER/WERD) over multiple scenarios, we follow the practice of Hendrycks & Dietterich (2019) and divide the metric by a measure of difficulty, i.e., by the (estimated) speech quality degradation. This adds weight to errors on "easy" scenarios (less quality degradation) and underweights errors on "harder" scenarios (more quality degradation) when computing averages. We refer to the difficulty normalized versions of WER/WERD as **Normalized WER/WERD** (NWER/NWERD). We estimate speech quality using DNSMOS (Reddy et al., 2019) and PESQ(Rix et al., 2001; Miao Wang & ananda seelan, 2022), which are models of human judgments of speech quality and predict Mean Opinion Scores (MOS, Rec 2018). PESQ uses various signal processing methods to predict MOS, while DNSMOS uses DNNs to do the same. To compute speech quality degradation we compute PESQ and DNSMOS for each clean and noisy recording multiplied by -1 (lower values indicate less degradation). Since we are only interested in the relative degradation between scenarios, we normalize the scores to have mean 50 and standard deviation 25.

**Note on usage:** We use NWERD for non-adversarial scenarios (1-5) but WERD for adversarial attacks because adversarial attacks are model-specific and thus DNSMOS/PESQ scores for adversarially perturbed audio will be different for each model, which will lead to a different normalization during NWERD computation and make comparisons difficult.

## 4 EVALUATION

We evaluate several recent ASR DNNs (§4.1) using SRB and analyze the results to uncover fine-grained differences in their robustness in various challenging scenarios. We further extend our analysis by measuring ASR model robustness for various sub-groups, namely English speech and non-English (Spanish) speech, and male and female speakers. Prior works (Liu et al., 2022; Veliche & Fung, 2023) observe that there is a disparity in transcription quality between subgroups. Our analysis augments these observations by showing that inter-group disparities in robustness may also exist, thus demonstrating the utility of SRB in the broader field of trustworthy AI.

## 4.1 MODELS

For English, we evaluate Whisper (Radford et al., 2023) large-v2, base, medium, small, and tiny (wsp-{lg,bs,md,sm,tn}), Wav2Vec-2.0 (Baevski et al., 2020) base, large, self-trained large (Xu et al., 2021), and Robust Wav2Vec (Likhomanenko et al., 2020) (w2v2-{bs,lg,lg-slf,lg-rob}), HuBERT (Hsu et al., 2021a) large and XL (hubt-{lg,xl}), Nvidia Canary (NVIDIA) (cnry-1b), Nvidia Parakeet RNN-T and CTC (NVIDIA) with 0.6B and 1.1B parameters (prkt-rnnt-{0.6,1.1}b, prkt-ctc-{0.6,1.1}b), MMS (Pratap et al., 2020) (mms-1b), Speech-T5 (Ao et al., 2022) (spch-t5), DeepSpeech (Amodei et al., 2016) (ds), and Speechbrain (Ravanelli et al., 2024) models with Conformer encoders, and transformer and RNN-T decoders. For Spanish speech, we evaluate mono-lingual Wav2Vec base Spanish (Wang et al., 2021b) (w2v2-bs-es), Wav2Vec XLSR Spanish (Conneau et al., 2020) (w2v2-lg-es), wsp-{lg,bs,tn}, and mms-1b. We used the Huggingface implementations where available, except ds (https://github.com/SeanNaren/deepspeech.pytorch). More details about the models are in Table 5.

## 4.2 ROBUSTNESS OF ASR MODELS

Table 1 presents the utility and robustness of a subset of English and Spanish ASR models under non-adversarial and adversarial scenarios. The subset was selected to exclude small and/or less accurate models. The results of the excluded models, however, are used in § 4.3.

| Lang | Model | clean (WER) | accent | audio proc | noise (env) | noise (white) | sFX | social (FF) | social (NF) | spatial | synth speech | AVG | Adv (UA) | Adv (US) | AVG |
|------|-------|------|--------|------------|-------------|---------------|-----|------|------|---------|--------------|-----|----------|----------|-----|
| | | | | | | | (NWERD) | | | | | | | (WERD) | |
| EN | wsp-lg | 8.0 | **11.2** | 12.7 | 3.1 | 2.9 | **2.8** | 40.9 | 34.9 | **4.5** | 4.4 | **13.0** | 6.2 | 53.6 | 29.9 |
| | prkt-rnnt-1.1b | **5.9** | 11.6 | **8.7** | 4.2 | **1.6** | 6.4 | 48.3 | 39.8 | 11.8 | **3.0** | 15.0 | 10.9 | 69.1 | 40.0 |
| | wsp-md | 7.9 | 12.4 | 27.8 | **2.8** | 3.3 | 3.5 | 41.8 | 35.5 | 5.2 | 6.1 | 15.4 | 3.7 | 56.6 | 30.1 |
| | prkt-ctc-1.1b | 6.0 | 16.9 | 10.3 | 4.2 | 3.2 | 9.6 | 44.6 | 35.0 | 13.5 | 5.9 | 15.9 | 8.4 | 71.2 | 39.8 |
| | cnry-1b | 6.0 | 13.9 | 18.2 | 4.4 | 2.7 | 15.0 | 45.7 | 36.7 | 15.3 | 5.7 | 17.5 | 14.8 | 61.7 | 38.3 |
| | w2v2-lg-slf | 7.7 | 41.0 | 30.7 | 13.1 | 17.6 | 20.3 | 69.5 | 67.6 | 26.3 | 15.2 | 33.5 | 7.0 | 41.0 | 24.0 |
| | hubt-xl | 8.4 | 38.9 | 29.1 | 15.5 | 16.2 | 20.8 | 69.5 | 71.2 | 26.4 | 13.5 | 33.5 | 13.9 | 36.8 | 25.3 |
| | wsp-bs | 9.6 | 30.1 | 88.8 | 8.7 | 9.6 | 22.5 | 63.9 | 47.8 | 17.9 | 12.6 | 33.6 | **2.7** | 88.5 | 45.6 |
| | w2v2-lg | 9.7 | 60.6 | 39.5 | 19.0 | 26.0 | 24.1 | 77.3 | 79.9 | 37.3 | 17.7 | 42.4 | 16.6 | **31.1** | **23.8** |
| ES | cnry-1b | **3.2** | 699.9 | 21.7 | 17.4 | 7.6 | 30.5 | - | - | 36.3 | 28.6 | 120.3 | 26.1 | 84.3 | 55.2 |
| | wsp-lg | 5.8 | **6.8** | **19.4** | **12.3** | **5.5** | **9.2** | - | - | **5.5** | **24.6** | **11.9** | 13.7 | 65.0 | 39.4 |
| | w2v2-lg-es | 6.8 | 31.6 | 31.8 | 30.1 | 20.5 | 40.7 | - | - | 89.0 | 104.1 | 49.7 | 33.9 | 71.0 | 52.4 |
| | wsp-bs | 14.8 | 62.0 | 133.2 | 43.1 | 25.8 | 60.4 | - | - | 58.1 | 87.8 | 67.2 | 19.5 | 159.5 | 89.5 |
| | mms-1b | 15.7 | 26.3 | 27.9 | 32.7 | 9.3 | 43.3 | - | - | 47.3 | 49.9 | 33.8 | **7.4** | 53.8 | 30.6 |
| | w2v2-bs-es | 25.7 | 45.6 | 55.9 | 44.9 | 25.0 | 58.4 | - | - | 103.3 | 152.7 | 69.4 | 10.0 | **33.8** | **21.9** |

Table 1: The utility and robustness of selected English and Spanish models (see Table 7 for more results). Utility is measured by WER of the models on clean speech. Robustness is measured by the NWERD on non-adversarially perturbed speech and WERD on adversarially perturbed speech. Adv (UA) refers to utterance agnostic attacks, while Adv (US) refers to utterance specific ones. The metrics are averaged over all datasets, perturbations, and severities in each category.

### 4.2.1 ROBUSTNESS IN NON-ADVERSARIAL SCENARIOS

**English Models:** In terms of average NWERD, we observe that wsp-lg emerges as the most robust model for non-adversarial scenarios, followed by prkt-rnnt-1.1b and wsp-md. Interestingly, cnry-1b, which is the top model on the Open ASR Leaderboard (OAL, Srivastav et al. 2023), ranks 5th on SRB. This result highlights the fact that SRB provides a more rigorous assessment of a model's robustness than existing benchmarks like OAL. We also see that SRB reveals subtle weaknesses and strengths of various models.

For instance, we see that wsp-lg and wsp-md are significantly more robust to special effects (sFX) and spatial acoustics than other models, including cnry-1b. To identify the specific types of sFX and spatial acoustic perturbations against which cnry-1b lacks robustness, we plot the WERD on each perturbation within these categories (Fig. 3) we find that cnry-1b is much more sensitive than its peers to echo, real room impulse responses, and speed and pitch modifications. This analysis demonstrates that SRB can evaluate the robustness of ASR models at multiple granularities and can pinpoint the weaknesses of a given model. Detailed results can be found in Figs. 9 and 11 in the appendix. Given that no data augmentation, other than SpecAugment (Park et al., 2019), was used to train Whisper (Radford et al., 2023), this indicates that Whisper was trained on data that may have included digital media like music or movie soundtracks, and speech recorded in diverse acoustic environments – settings that may not be sufficiently represented in public data sources. Curating such diverse datasets is a promising direction for future work.

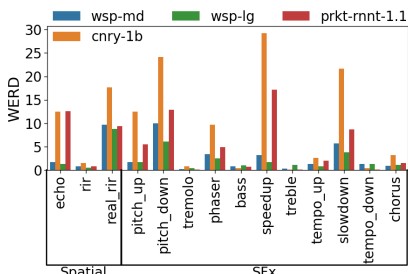

Figure 3: WERD of cnry-1b, wsp-md and wsp-lg on perturbations in the spatial acoustics and special effects categories.

We also note that despite being pre-trained on 60K hours of speech, Wav2Vec and Hubert models severely lack robustness. Particularly concerning is their weak performance on accented speech, social scenarios and spatial acoustics, which models are very likely to encounter in the real world.

***Takeaways:*** *(1) Despite topping the Open ASR Leaderboard, cnry-1b is significantly less robust than wsp-lg, which is ranked 10 on OAL. cnry-1b particularly lacks robustness to special effects and spatial acoustics. (2) Wav2Vec variants struggle against accented speech and social settings, thus, may not be suitable when users have diverse accents.*

**Spanish Models:**We observe that wsp-lg is the most robust model against non-adversarial perturbations by some margin. We notice that all models, except wsp-lg, struggle against accented speech and yield high NWERS. cnry-1b is particularly, weak against accented speech with an NWERD of 700% (WERD=205%). Apart from accented speech, cnry-1b is quite robust on all other categories of non-adversarial perturbations. mms-1b is also fairly robust and, unlike other models, its NWERD does not vary erratically from one category to another.

### 4.2.2 Robustness in Adversarial Scenarios

**English models:**We observe that w2v2-lg achieves the lowest WERD and thus is the most robust model against utterance-specific adversarial attacks. Interestingly, while Wav2Vec models exhibited mediocre robustness to non-adversarial perturbations, they are more robust to utterance-specific attacks, than Whisper, Canary, and Parakeet, which were the most robust on non-adversarial perturbations. We also note from Fig. 8 (in Appendix) that most Wav2Vec models are considerably more robust to attacks against TEDLIUM than against LibriSpeech, and the opposite is true for whisper and Canary models. Under utterance-agnostic attacks, the most robust models are mms-1b, wsp-bs, and wsp-sm. It is interesting to note that the smaller variants of Whisper limit the generalizability across utterances of the adversarial perturbations to a greater extent than their larger counterparts.

*Takeaway: Wav2Vec models are most robust to adversarial attacks; Models that are most robust to non-adversarial perturbations, are mediocre against adversarial perturbations; Canary and Parakeet models are highly vulnerable to utterance specific attacks.*

**Spanish models:**On Spanish, w2v2-bs-es is the most adversarially robust model. Generally, we observe that Wav2Vec models exhibit better robustness than Whisper and Canary under both utterance-agnostic and utterance-specific perturbations. This is similar to the trends observed in English speech (Fig. 8c). Detailed results can be found in Fig. 10 in the appendix.

*Takeaway: General trends similar to English but WERD is higher when Spanish speech is attacked.*

### 4.3 Correlates of Robustness

To glean insights that can inform future work, we have conducted the following analysis to model attributes that yield robust models. Specifically, we examine the impact of model size, architecture and accuracy, as well as training dataset size on robustness.

To determine if the prevailing practice of training DNNs with more parameters on larger datasets is yielding improvements in robustness, we use robust linear regression to fit a line to WERD vs. number of model parameters/size of the training data for the candidate models in Figs. 5a and 5b, respectively. Increasing model size is correlated with improved robustness (lower WERD).

To further isolate the impact of the model size we plot the NWERD of models from the same family in Fig. 4, which have similar architectures and training datasets. We note that larger models are more robust in the Whisper, Parakeet and Wav2Vec-2.0 families, but, surprisingly, not in the HuBert family.

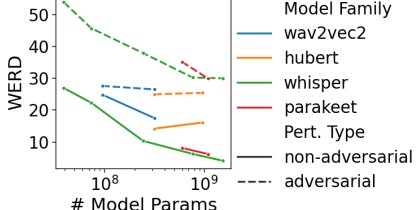

Figure 4: NWERD lineplot with non-adversarial and adversarial perturbations, three families of models.

Next, we consider the model architectures. The architectures of the models used in this paper can be divided in to three categories: sequence-to-sequence (seq2seq) models like Whisper and Canary, encoder only models trained with CTC loss (Graves et al., 2006) like the Wav2Vec family, and RNN-T models which are capable of streaming such as some variants of Parakeet. From Fig. 5e we see that in terms of non adversarial robustness RNN-T models outperform seq2seq and CTC models, but in terms of adversarial robustness CTC models achieve the lowest WERD.

We also measure the robustness-utility trade-off by plotting WERD and NWERD for adversarial and non-adversarial perturbations, respectively, against WER on clean data in Figs. 5c and 5d. We observe that in both cases the relationship is positive, i.e. more accurate models tend to be more robust, however, the relationship between WERD on adversarial perturbations and clean WER is much weaker.

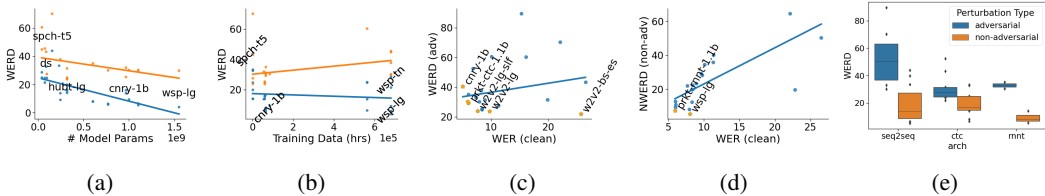

(a)     (b)     (c)     (d)     (e)

Figure 5: (a & b) WERD for all models with robust regression fitted line on non-adversarial (blue) and adversarial (orange) perturbations, plotted against (a) number of parameters, and (b) hours of training data. (c % d) WERD and NWER on adversarial and non-adversarial perturbations are plotted against WER to illustrate the robustness-utility trade off. Pareto optimal points are highlighted. (e) Boxplot of WERD for models having various architectures.

Finally, we measure the impact of training data size and, find that increasing training data appears to have only a minor influence on robustness (Fig. 5b).

***Takeaway:*** *(1) Larger models tend to be more robust, while smaller models, even if they are trained on large datasets, are less robust. This runs somewhat counter to the prevailing wisdom (Radford et al., 2023; Likhomanenko et al., 2020). (2) CTC models are more robust than seq2seq models to adversarial attack, but less robust than seq2seq and RNN-T models on non-adversarial perturbations. (3) Utility and robustness are positively correlated, but correlation is weaker for adversarial robustness.*

## 4.4 DISPARITY IN ROBUSTNESS ACROSS POPULATION SUB-GROUPS

In the preceding analysis, we considered robustness aggregated over the entire population (i.e., dataset). However, populations are generally not homogeneous, and, thus, the robustness of the model may differ on various population sub-groups. Prior works have commonly analyzed sub-group fairness of ASR models by comparing the overall WER for each sub-group on a benchmark dataset (Koenecke et al., 2020). It is possible that models that are fair *on average*, may not be fair under certain conditions. In the following, we use SRB to uncover and analyze the disparities in the models' robustness across four sub-groups: English and Spanish speech, and male and female speakers. We find that disparities indeed exist, with multi-lingual models generally being more robust for English than Spanish (Fig. 6), and most models being less robust for females than males.

### 4.4.1 DISPARITY IN ROBUSTNESS ACROSS LANGUAGES IN MULTI-LINGUAL MODELS

We compare the robustness exhibited by multi-lingual models, wsp-lg, wsp-bs, cnry-1b and mms-1b on English and Spanish. The WERD of these models on both languages is presented in Fig. 6. We observe that Whisper models achieve lower WERD on English speech than on Spanish on almost all perturbation categories, while cnry-1b and mms-1b achieve similar WERD on some categories. We also note that the difference in WERD on some common perturbation categories, like environmental noise, and spatial acoustics, is much greater for wsp-lg than for cnry-1b.

***Takeaway:*** *Multilingual models are more robust on English than Spanish; cnry-1b and wsp-lg most robust on both languages; adversarial robustness results follow the same trend as English.*

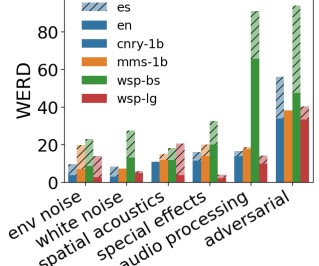

Figure 6: Comparing the robustness of multi-lingual on English (solid) and Spanish (hatched).

### 4.4.2 DISPARITY IN ROBUSTNESS ACROSS GENDERS

To measure the disparity in transcription quality across genders (males/females), we compute the log of the ratio of the WERs of the ASR model on female and male speakers. We call this measure the Log WER Ratio (LWERR). A positive value of LWERR indicates that the model is biased against females and a negative value indicates that the model is biased against males.

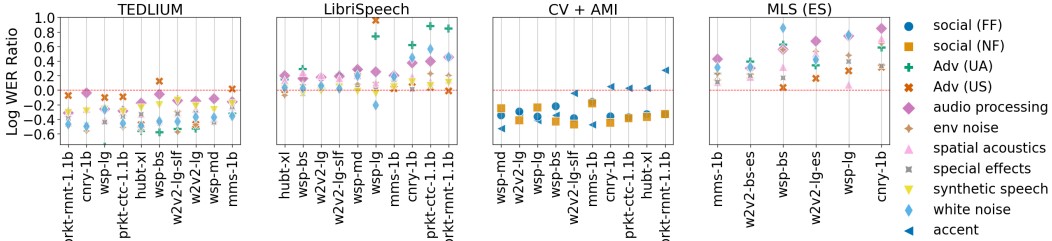

Figure 7: Log WER Ratio for various TEDLIUM, LibriSpeech, Common Voice (CV) and AMI, and Spanish Multilingual Librispeech (MLS (ES)). WERs are averaged across severity levels and individual augmentations within each category before computing the Log WER Ratio.

The LWERR for each dataset is shown in Fig. 7. We note that, on average, the models are biased against females on LibriSpeech and Spanish Multilingual Librispeech (MLS-ES), and against males on TEDLIUM, Common Voice and AMI. The bias is most prominent in MLS-ES, where cnry-1b seems to be yielding the highest disparities among genders. We also note that adversarial perturbations cause the WER of wsp-lg to increase significantly more for females than males in LibriSpeech. This is interesting because adversarial perturbations do not target a specific part of the spectrum and thus should not impact one gender more than the other.

***Takeaway:*** *Models are more robust for males on some datasets, and females on other datasets suggesting that used data require further examination; adversarial attacks increase WER of Whisper variants for females more than males; multilingual models, particularly* cnry-1b*, are more biased against females when transcribing Spanish.*

## 5 Limitations

Despite the comprehensive design of SRB, there are limitations to consider. Firstly, while SRB includes a diverse set of datasets, distortions and adversarial attacks, it may not encompass all possible real-world scenarios. Additionally, the benchmark has been tested for now on English and Spanish languages, which may not capture the robustness challenges faced by ASR models in other languages and dialects. While these limitations potentially affect the generalizability of the results obtained with SRB, it extensibility allows users to easily incorporate additional datasets and models.

## 6 Conclusion

We propose SRB, a comprehensive benchmark designed to standardize the robustness evaluations of Automatic Speech Recognition (ASR) models. We demonstrate the utility of SRB in evaluating the robustness of ASR models via several concrete examples, as well as its potential to facilitate evaluations of other aspects of trustworthy AI, like fairness. We believe that SRB will enable rigorous robustness evaluations of ASR models in a highly standardized manner, allowing easy comparisons between existing and new approaches. To further facilitate robustness evaluations for researchers and model developers, we release transformed test sets in English and Spanish. We anticipate that this will make robustness evaluations more prevalent and encourage model developers to consider robustness as a key metric for improvement.

## 7 Acknowledgments

Funded by the European Union's Horizon 2020 RIA ELOQUENCE project (Grant Agreement No. 101135916), by the Spanish Project 6G-RIEMANN (Grant Agreement No. 2022/0005420), and partially also by the European Lighthouse on Safe and Secure AI (ELSA) from the European Union's Horizon Europe program under grant agreement No 101070617. Views and opinions expressed are however those of the author(s) only and do not necessarily reflect those of the European Union or European Commission-EU. Neither the European Union nor the granting authority can be held responsible for them.

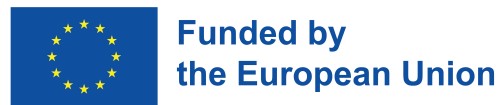

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

## A    PERTURBATION GENERATION/APPLICATION PROCEDURE

Below, we provide further details the perturbations that make up `Speech Robust Bench`. Table 2 shows the parameters for each perturbation and Table 3 shows the normalized DNSMOS and PESQ scores for each perturbation.

GAUSSIAN NOISE:    A noise vector of the same length as the audio signal is sample from a standard normal distribution, scaled such that its magnitude corresponds to a specific SNR, and then added to the audio signal. We use `torchaudio.function.add_noise` to add the noise to the speech at a given SNR.

ENVIRONMENTAL NOISE:    We use the recordings of environmental noises from the test/eval subsets of ESC-50 (Piczak, 2015), MS-SNSD (Reddy et al., 2019), MUSAN (Snyder et al., 2015) and WHAM (Wichern et al., 2019). We create a separate perturbed version of the clean data using each of these noise datasets. To do so, for each test utterance we sample a random environmental noise and add it to the audio signal at the specified SNR. We clip the noise if it is longer than the speech, and repeat it if it is shorter than the speech. We use `torchaudio.function.add_noise` to add the noise to the speech at a given SNR.

ROOM IMPULSE RESPONSE:    The simulated and real RIRs from (Ko et al., 2017) are applied to clean recordings. As a measure of intensity, RT60 is estimated for the simulated RIRs using Sabine's formula with the room dimensions and absorption coefficient provided in the dataset. For the real RIRs, we compute the SRMR (Santos & Falk, 2014) using the implementation from `https://github.com/aliutkus/speechmetrics/tree/master`. The severity is defined in increasing RT60s for the synthetic RIRs, and decreasing SRMR for the real RIRs. Table 2 shows the average RT60/SRMS in each severity level. During evaluation, a random RIR having the given severity level is sampled for each test recording.

RESAMPLING, SPEED, PITCH, AND GAIN PERTURBATIONS:    The resampling speed, pitch, and gain perturbations were applied using the `Resample Speed`, `PitchShift` and `Vol` transforms from `torchaudio`.

| Category | Perturbation | Sev 1 | Sev 2 | Sev 3 | Sev 4 |
|---|---|---|---|---|---|
| Environment | Gaussian Noise | 30 dB | 20 dB | 10 dB | 0 dB |
| | Environmental Noise | 30 dB | 20 dB | 10 dB | 0 dB |
| | Music | 30 dB | 20 dB | 10 dB | 0 dB |
| | Crosstalk | 30 dB | 20 dB | 10 dB | 0 dB |
| Spatial Acoustics | RIR | 0.27s | 0.58s | 0.99s | 1.33s |
| | Real RIR | 9.1 | 7.1 | 4.1 | 1.8 |
| | Echo (delay) | 125 ms | 250 ms | 500 ms | 1000 ms |
| Special Effects | Bass (gain) | 20 | 30 | 40 | 50 |
| | Treble (gain) | 10 | 23 | 36 | 50 |
| | Phaser (decay) | 0.3 s | 0.5 s | 0.7 s | 0.9 s |
| | tempo-up | 1.25x | 1.5x | 1.75x | 2x |
| | tempo-down | 0.875x | 0.75 | 0.625x | 0.5x |
| | Speed-up | 1.25x | 1.5x | 1.75x | 2x |
| | Slow-down | 0.875x | 0.75 | 0.625x | 0.5x |
| | Pitch Step-up | 0.25 oct | 0.5 oct | 0.75 oct | 1 oct |
| | Pitch Step-down | 0.25 oct | 0.5 oct | 0.75 oct | 1 oct |
| | Chorus (delay) | 30 | 50 | 70 | 90 |
| | tremolo (depth) | 50 | 66 | 83 | 100 |
| Audio Processing | Resampling | 0.75x | 0.5x | 0.25x | 0.125x |
| | Gain (factor) | 10x | 20x | 30x | 40x |
| | Low-pass filter | 4 kHz | 2833 kHz | 1666 kHz | 500 kHz |
| | High-pass filter | 500 kHz | 1333 kHz | 2166 kHz | 3000 kHz |
| Adversarial | PGD Attack | 40 dB | 30 dB | 20 dB | 10dB |
| | Utterance Agnostic Attack | 40 dB | 30 dB | 20 dB | 10dB |

Table 2: The parameters defining the various severity levels of the perturbations used in the proposed benchmark.

OTHER SPECIAL EFFECTS: These effects are applied via SoX filters of the same name. We used `torchaudio.sox_effects.apply_effects_tensor` to apply these filters to the audio. The args for each filter are as follows:

- `echo 0.8 0.9 <delay> 0.3`

- `phaser 0.6 0.8 3 <decay> 2 "-t"`

- `Tempo <factor> 30`

- `sinc <lo-freq>`

- `sinc 0-<hi-freq>`

- `tremolo 20 <depth>`

- `treble <gain>`

- `bass <gain>`

- `chorus 0.9 0.9 <delay> 0.4 0.25 2 -t {<delay>+10} 0.3 0.4 2 -s`

VOICE CONVERSION We use use YourTTS (Casanova et al., 2022) from Coqui.ai[5] to synthesize audio from textual transcripts in a given speaker's style. The transcripts from the test clean subset of LibriSpeech are used. The target speakers are drawn from the VCTK corpus (Yamagishi et al., 2019), which contains accented speech from 12 accents. For each transcript a random speaker is chosen to synthesize the audio.

---

[5]https://github.com/coqui-ai/TTS

| Metric → Scenario | AVG | | | | normalized DNSMOS | | | | normalized PESQ | | | |
|---|---|---|---|---|---|---|---|---|---|---|---|---|
| clean | 23.1 | | | | 23.1 | | | | | | | |
| accent (en) | 33.1 | | | | 33.1 | | | | | | | |
| accent (es) | 29.3 | | | | 29.3 | | | | | | | |
| social (chime, FF) | 102.2 | | | | 102.2 | | | | | | | |
| social (ami, FF) | 85.9 | | | | 85.9 | | | | | | | |
| social (chime, NF) | 80.1 | | | | 80.1 | | | | | | | |
| social (ami, NF) | 37.3 | | | | 37.3 | | | | | | | |
| Augmentation/Severity | 1 | 2 | 3 | 4 | 1 | 2 | 3 | 4 | 1 | 2 | 3 | 4 |
| bass | 19.1 | 23.9 | 36.0 | 56.4 | 23.9 | 28.9 | 39.6 | 58.4 | 11.4 | 14.3 | 30.3 | 54.6 |
| chorus | 40.1 | 49.3 | 55.5 | 57.0 | 31.3 | 41.3 | 48.7 | 49.9 | 63.8 | 71.2 | 74.3 | 75.8 |
| crosstalk | 22.9 | 38.9 | 53.1 | 59.9 | 24.7 | 33.1 | 38.2 | 41.0 | 22.2 | 46.1 | 70.1 | 81.6 |
| echo | 54.9 | 54.2 | 53.6 | 51.4 | 40.6 | 37.8 | 37.6 | 36.5 | 71.3 | 72.7 | 71.7 | 67.9 |
| env noise (MS-SNSD) | 51.4 | 62.4 | 77.0 | 89.6 | 53.5 | 61.0 | 74.6 | 93.5 | 39.6 | 58.7 | 76.7 | 84.2 |
| env noise (ESC50) | 26.7 | 41.7 | 58.4 | 73.8 | 39.2 | 45.3 | 55.0 | 73.2 | 20.0 | 43.1 | 66.0 | 79.8 |
| env noise (MUSAN) | 24.9 | 43.0 | 63.0 | 76.4 | 26.3 | 38.7 | 57.1 | 73.1 | 24.8 | 48.7 | 70.4 | 81.0 |
| env noise (WHAM) | 23.0 | 46.2 | 74.2 | 93.3 | 23.7 | 41.2 | 72.3 | 101.7 | 23.1 | 52.0 | 76.6 | 85.4 |
| gain | 50.8 | 69.9 | 77.6 | 81.8 | 46.7 | 67.8 | 78.3 | 84.6 | 61.3 | 76.0 | 80.0 | 81.7 |
| gaussian noise | 53.2 | 76.6 | 91.7 | 82.7 | 72.1 | 89.5 | 103.8 | 119.6 | 42.1 | 69.2 | 83.0 | 66.0 |
| highpass | 40.9 | 56.2 | 68.5 | 78.5 | 35.8 | 45.1 | 66.7 | 83.0 | 46.1 | 68.3 | 71.6 | 74.9 |
| lowpass | 33.8 | 37.9 | 51.6 | 79.1 | 48.9 | 48.4 | 64.3 | 100.1 | 20.3 | 29.2 | 40.4 | 58.4 |
| music | 22.9 | 43.8 | 66.7 | 79.9 | 26.9 | 43.0 | 64.0 | 78.6 | 20.0 | 45.3 | 70.1 | 81.9 |
| phaser | 15.6 | 32.9 | 60.7 | 80.6 | 22.8 | 36.2 | 59.8 | 78.6 | 10.6 | 31.9 | 63.6 | 83.5 |
| pitch down | 61.8 | 68.2 | 63.8 | 84.4 | 39.6 | 50.9 | 67.3 | 82.8 | 85.9 | 86.5 | 68.1 | 86.7 |
| pitch up | 58.9 | 62.1 | 65.2 | 66.0 | 33.7 | 39.6 | 45.7 | 47.5 | 85.9 | 86.4 | 86.5 | 86.4 |
| real rir | 39.4 | 54.6 | 69.9 | 85.3 | 35.7 | 46.5 | 61.7 | 89.2 | 43.1 | 62.7 | 78.0 | 81.4 |
| resample | 14.9 | 28.0 | 49.9 | 64.2 | 25.2 | 43.7 | 63.3 | 77.4 | 6.7 | 18.0 | 38.1 | 52.5 |
| rir | 51.1 | 64.0 | 69.3 | 68.9 | 42.0 | 58.1 | 66.2 | 66.5 | 65.0 | 74.6 | 78.1 | 78.1 |
| slowdown | 51.4 | 57.8 | 65.2 | 74.0 | 18.9 | 31.2 | 45.3 | 68.6 | 85.9 | 86.1 | 86.0 | 86.3 |
| speedup | 52.2 | 59.6 | 67.2 | 73.8 | 23.1 | 37.6 | 52.6 | 65.5 | 83.7 | 83.6 | 83.3 | 83.0 |
| tempo down | 49.4 | 52.6 | 55.3 | 51.2 | 19.8 | 23.0 | 28.2 | 36.7 | 81.4 | 84.7 | 85.3 | 85.6 |
| tempo up | 51.0 | 57.9 | 64.1 | 70.6 | 25.7 | 36.1 | 47.8 | 60.1 | 79.0 | 82.2 | 82.4 | 82.4 |
| treble | 12.4 | 22.4 | 41.6 | 63.6 | 20.8 | 31.2 | 44.8 | 62.3 | 2.1 | 12.1 | 42.2 | 72.5 |
| tremolo | 17.7 | 29.9 | 60.2 | 100.2 | 24.5 | 38.8 | 73.9 | 113.7 | 9.6 | 17.6 | 36.9 | 75.6 |
| synthetic (es, Bark) | 30.7 | - | - | - | 30.7 | - | - | - | - | - | - | - |
| synthetic (en, yourTTS) | 50.3 | - | - | - | 17.1 | - | - | - | 83.6 | - | - | - |

Table 3: Normalized DNSMOS and PESQ score for each perturbation.

| Name | Subset | Hours | Utterances | Speakers | Male/Female |
|---|---|---|---|---|---|
| LibriSpeech | test-clean | 5.4 | 2620 | 40 | 20/20 |
| TEDLIUM 3 | test | 3.76 | 1155 | 16 | 10/6 |
| Multi-Lingual LibriSpeech (es) | test | 10 | 2385 | 20 | 10/10 |
| CHiME-6 | eval | 5.25 | 13000 | 8 | - |
| AMI | test | 7.35 | 13168 | 16 | 8/8 |
| ESC-50 | - | 2.78 | 2000 | - | - |
| MUSAN | - | 108.5 | 2016 | - | - |
| WHAM! | noise-test | 9 | 3000 | - | - |
| MS-SNSD | noise-test | 0.7 | 51 | - | - |

Table 4: Distributional statistics of speech (top) and noise (bottom) datasets used in SRB.

CROSSTALK AND MUSIC We use crosstalk and music audios from MUSAN (Snyder et al., 2015). We use `torchaudio.function.add_noise` to add the noise to the speech at a given SNR.

ACCENTS    We select a subset of audios from the test set of Common Voice 17. The selected audios satisfied the following criteria: (1) the speaker's accent must be present in the metadata, (2) the accent must not be American, (3) should be clean. The last criterion is satisfied if the DNSMOS (Reddy et al., 2020) score of the recording is at least 3.4. The resulting subset contains 640 recordings. The most popular accent in this set is South Asian (India, Pakistan, Sri Lanka) ($\tilde{2}5\%$), followed by British English ($\tilde{2}5\%$).

INTER-PERSONAL COMMUNICATIONS    We CHiME-6 (Barker et al., 2017) and AMI (Kraaij et al., 2005) to obtain recordings of people in social scenarios (dinner party and meetings). In both these datasets, the speakers are recorded through lapel microphone and a room microphone resulting in near and far field recordings. We use both types of recordings and show separate results for them. We remove recordings that contain less than three words since they are often fillers.

UTTARANCE SPECIFIC ADVERSARIAL ATTACK:    The utterance specific adversarial perturbations are computed using the *untargeted* PGD adversarial attack implemented in `robust_speech` package (Olivier & Raj, 2022). The attack is computed as follows. First, the maximum possible L2 norm of the noise is determined by solving the equation for SNR for the norm of the noise as follows.

$$\text{SNR} = 20 \log_{10} \left( \frac{||x||_2}{||\delta||_2} \right) \tag{1}$$

$$\epsilon_{\text{SNR}} = ||\delta||_2 = 10^{-\frac{\text{SNR}}{20}} ||x||_2, \tag{2}$$

where $\delta$ is the noise, $x$ is the audio signal and SNR is the upper bound on the SNR in the final signal. Then, we follow the approach of (Madry et al., 2018) and optimize $\delta$ using Projected Gradient Descent (PGD) to maximize the divergence between the true and predicted transcriptions. Formally stated, the attack performs the following optimization:

$$\delta = \max_{\hat{\delta}: \left\| \hat{\delta} \right\|_2 \leq \epsilon_{\text{SNR}}} L_M(x + \hat{\delta}, y), \tag{3}$$

where $L_M$ is the loss function used to train the ASR model, $M$, such as CTCLoss or NLLLoss.

UTTERANCE AGNOSTIC ADVERSARIAL ATTACK:    We use the method of (Neekhara et al., 2019), as implemented in `robust_speech` package (Olivier & Raj, 2022), to compute utterance agnostic adversarial perturbations. The main difference between the universal attack and the PGD attack is that the latter computes a perturbation vector for each input, whereas the former computes a single perturbation that is expected to successfully attack any input to the model.

Formally, given a ASR model, $M$, and a development speech dataset, $\mathcal{X}^{\text{dev}}$ let $\mathcal{X}_\delta^{\text{dev}} = \{x + \delta | x \in \mathcal{X}^{\text{dev}}\}$ be the same dataset under additive perturbation $\delta$, and let $M(\mathcal{X}^{\text{dev}})$ and $M(\mathcal{X}_\delta^{\text{dev}}$ be the transcripts predicted by $M$ for $\mathcal{X}^{\text{dev}}$ and $\mathcal{X}_\delta^{\text{dev}}$. The utterance agnostic attack uses PGD to optimize $\delta$ such that $||\delta||_\infty \leq \epsilon$ and the Character-Error Rate (CER) (see § B) between $M(\mathcal{X}^{\text{dev}})$ and $M(\mathcal{X}_\delta^{\text{dev}})$ is at least $t$, i.e. $\text{CER}^M(\mathcal{X}_\delta^{\text{dev}}, M(\mathcal{X}^{\text{dev}})) \geq t$ (using the notation from § B). Similar to the utterance-specific attack, the value of $\epsilon$ is determined by the maximum allowable SNR using eq.(2), except that $\ell_\infty$ norms are used instead of $\ell_2$ norms. The full algorithm is described in Algorith 1.

Once we compute the perturbation we add it to the test audios ($\mathcal{X}^{\text{test}}$) at the specified SNR using `torchaudio.function.add_noise`. For LibriSpeech, we use 500 utterances from test-dev as $\mathcal{X}^{\text{dev}}$ and test-clean as $\mathcal{X}^{\text{dev}}$. For TEDLIUM, we use the full dev and test sets as $\mathcal{X}^{\text{dev}}$ and $\mathcal{X}^{\text{test}}$. For Multi-Lingual LibriSpeech, we use 500 utterances from the dev set in the relevant language as $\mathcal{X}^{\text{dev}}$ and the full test set of the same language as $\mathcal{X}^{\text{test}}$.

# B    ADDITIONAL DEFINITIONS

WORD ERROR RATE:    As noted in the main text, we use word error rate (WER) as a basic measure for quantifying performance of the models. Following the common practice from ASR literature, the WER is computed as the word-level edit distance between the reference and the predicted transcripts, normalized by the length of the reference. The edit distance is computed as the total number of word substitutions, deletions, and additions required to transform the reference transcript into the predicted

---

**Algorithm 1** Utterance Agnostic Attack Algorithm

---

**Require:** Speech data $\mathcal{X}^{\text{dev}}$ and , ASR model $M$ and its loss function, $L_M$ (e.g. CTCLoss, NLLLoss, etc.), allowed SNR $s$, learning rate, $\alpha$, max epochs $e_{max}$, max iterations per sample $i_{max}$, target attack success rate, $t_{sr}$, target Character-Error Rate (CER) (see § B), $t_{cer}$

    **procedure** CER($a, b$)
        **return** EditDistance$(a, b)/len(b)$                        ▷ $a$ and $b$ are character sequences
    **end procedure**
    **procedure** SUCCESSRATE($\mathcal{X}$)
        **return** $\sum_{x \in \mathcal{X}} I[CER(M(x+v), M(x)) > t_{cer}]$
    **end procedure**
    **procedure** SNRTONORM($x$, SNR)
        **return** $10^{-\frac{\text{SNR}}{20}} ||x||_\infty$
    **end procedure**
    $\epsilon \leftarrow \sum_{x \in \mathcal{X}^{\text{dev}}} SNRToNorm(x, s)/|\mathcal{X}^{\text{dev}}|$
    $v \leftarrow 0$
    $e \leftarrow 0$
    **while** $SuccessRate < t_{sr}$ and $e < e_{max}$ **do**
        **for** $(x, y) \in \mathcal{X}^{\text{dev}}$ **do**                    ▷ $x$ is the audio, $y$ is the transcript
            $i \leftarrow 0$
            $r \leftarrow 0$
            **while** $CER(M(x+v+r), M(x)) > t_{cer}$ and $i < i_{max}$ **do**
                $\Delta_r \leftarrow \alpha sign(\nabla_r 0.5 \left\|r\right\|_2 - L_M(x+v+r, y))$
                $r \leftarrow clip_\epsilon\{r - \Delta_r + v\} - v$
                $i \leftarrow i + 1$
            **end while**
            $v \leftarrow clip_\epsilon\{r + v\}$
        **end for**
        $e \leftarrow e + 1$
    **end while**

---

transcript. We remove all punctuation from both the reference and predicted transcripts, and convert to lower case before computing WER.

When WER is computed over multiple pairs of predicted and reference transcripts, it is common practice to sum the number of substitutions, deletions, and additions for all the pairs, and divide by the sum of the lengths of the reference transcripts. Formally, this can be written as:

$$WER^M(\mathcal{X}, \mathcal{R}) := 100 \frac{\sum_{x \in \mathcal{X}, r \in \mathcal{R}} ED(M(x), r)}{\sum_{r \in \mathcal{R}} |r|}\%, \tag{4}$$

where $ED$ computes the edit distance.

The Character Error Rate (CER) can also be defined similarly, by using the character-level edit distance in the above equation.

For quantifying differences between (binary) genders, we measure the disparity in prediction accuracy across males and females by the Log WER Ratio (LWERR). Formally,

$$LWERR := \log_2 \frac{WER^M(\mathcal{X}_f, \mathcal{R}_f)}{WER^M(\mathcal{X}_m, \mathcal{R}_m)}, \tag{5}$$

where the $(\mathcal{X}_f, \mathcal{R}_f)$ and $(\mathcal{X}_m, \mathcal{R}_m)$ represent the subsets of utterances by females and males respectively.

FAIRNESS THROUGH ROBUSTNESS: In this work, we use SRB to conduct a fairness assessment based on robustness disparities across population subgroups (English vs. Spanish speech; male vs. female speakers). Most of the state of the art quantifies fairness in terms of predictive performance disparities. This occurs in the domain of fairness in ASR systems (Liu et al., 2022; Veliche & Fung, 2023; Koenecke et al., 2020) and similar prevalence is found in other domains (Julia Angwin; Solans Noguero et al., 2023). However, previous work in the domain of ASR also considered robustness disparities as an alternative fairness notion (Nanda et al., 2021).

Moreover, we argue that considering the dimension of robustness could give better sense of the expected disparities that could be observed when deployed in the wild, in the presence of diverse noisy conditions.

## C  MODELS

Table 5 provides a summary of the models used in our evaluations. The model names correspond to the names of their pretrained checkpoints in the Huggingface library (`https://huggingface.co/models`). The abbreviations of these names are in the parentheses after them. Some of the unilingual models are pre-trained on multilingual data but are fine-tuned on only one language and thus can not transcribe any other language. Multilingual models have been pre-trained and fine-tuned on multiple languages so the same DNN can transcribe several languages. The WER of multilingual models is presented as English/Spanish.

## D  FINE GRAINED ANALYSES

The following figures present fine-grained analyses of robustness. These figures may be referenced by the main text but were not included in the main body in the interest of space. Figure 8 provides an overview of the accuracy and robustness of the various models. Figures 9, and 10 present the breakdown by perturbation of the robustness of models on English and Spanish, respectively. Figure 11 presents a breakdown of robustness by severity.

## E  COMPUTE RESOURCES

The experiments were performed on the Bridges-2 cluster at the Pittsburgh Supercomputing Center. This cluster contains 200 32G and 16G Nvidia V-100, which were used for these experiments.

| Lang | Model | Abrv. | Params (M) | Data (hrs) | WER |
|------|-------|-------|-----------|-----------|-----|
| EN | canary-1b (NVIDIA) | cnry-1b | 1,000.0 | 85,000 | 6.0 |
| | parakeet-ctc-0.6b (NVIDIA) | prkt-ctc-0.6b | 600.0 | 64,000.0 | 6.1 |
| | parakeet-ctc-1.1b (NVIDIA) | prkt-ctc-1.1b | 1,100.0 | 64,000.0 | 6.0 |
| | parakeet-rnnt-0.6b (NVIDIA) | prkt-rnnt-0.6b | 600.0 | 64,000.0 | 6.0 |
| | parakeet-rnnt-1.1b (NVIDIA) | prkt-rnnt-1.1b | 1,100.0 | 64,000.0 | 5.9 |
| | deepspeech(Amodei et al., 2016) | ds | 86.0 | 960 | 26.5 |
| | hubert-large-ls960-ft(Hsu et al., 2021a) | hubt-lg | 317.0 | 60,000 | 8.4 |
| | hubert-xlarge-ls960-ft(Hsu et al., 2021a) | hubt-xl | 964.0 | 60,000 | 8.4 |
| | mms-1b-fl102 (Pratap et al., 2023) | mms-1b | 964.6 | 55,000 | 22.8 |
| | speecht5 asr(Ao et al., 2022) | spch-t5 | 154.6 | 960 | 22.1 |
| | wav2vec2-base-960h (Baevski et al., 2020) | w2v2-bs | 95.0 | 960 | 11.3 |
| | wav2vec2-large-960h (Baevski et al., 2020) | w2v2-lg | 317.0 | 960 | 9.7 |
| | wav2vec2-large-960h-lv60-self (Xu et al., 2021) | w2v2-lg-slf | 317.0 | 60,000 | 7.7 |
| | wav2vec2-large-robust-ft-libri-960h (Hsu et al., 2021b) | w2v2-lg-rob | 317.0 | 63,000 | 8.9 |
| | whisper-base (Radford et al., 2023) | wsp-bs | 74.0 | 680,000 | 9.6 |
| | whisper-base.en (Radford et al., 2023) | wsp-bs.en | 74.0 | 563,000 | 5.1 |
| | whisper-large-v2 (Radford et al., 2023) | wsp-lg | 1,550.0 | 680,000 | 8.0 |
| | whisper-medium (Radford et al., 2023) | wsp-md | 769.0 | 680,000 | 7.9 |
| | whisper-medium.en (Radford et al., 2023) | wsp-md.en | 769.0 | 563,000 | 4.1 |
| | whisper-small (Radford et al., 2023) | wsp-sm | 244.0 | 680,000 | 8.3 |
| | whisper-small.en (Radford et al., 2023) | wsp-sm.en | 244.0 | 563,000 | 4.0 |
| | whisper-tiny (Radford et al., 2023) | wsp-tn | 39.0 | 680,000 | 11.3 |
| | whisper-tiny.en (Radford et al., 2023) | wsp-tn.en | 39.0 | 563,000 | 10.1 |
| ES | canary-1b (NVIDIA) | cnry-1b | 1,000.0 | 85,000 | 3.2 |
| | mms-1b-fl102 (Pratap et al., 2023) | mms-1b | 964.6 | 55,000 | 15.7 |
| | wav2vec2-base-10k-voxpopuli-ft-es (Wang et al., 2021b) | w2v2-bs-es | 94.4 | 10,116 | 25.7 |
| | wav2vec2-large-xlsr-53-spanish (Conneau et al., 2020) | w2v2-lg-es | 315.4 | 54,350 | 6.8 |
| | whisper-base (Radford et al., 2023) | wsp-bs | 74.0 | 680,000 | 14.8 |
| | whisper-large-v2 (Radford et al., 2023) | wsp-lg | 1,550.0 | 680,000 | 5.8 |
| | whisper-tiny (Radford et al., 2023) | wsp-tn | 39.0 | 680,000 | 23.3 |

Table 5: Models used in our evaluations.

| Dataset | License |
|---------|---------|
| LibriSpeech | CC-BY-4.0 |
| Multilingual Librispeech | CC BY 4.0 |
| TEDLIUM | CC-BY-NC-ND 3.0 |
| AMI | CC-BY-4.0 |
| Common Voice | CC0-1.0 |
| CHiME | CC BY-SA 4.0 |

Table 6: Licenses of each of the considered datasets in SRB

## F    DATASET LICENSES

The licenses of each of the considered datasets are described in Table 6.

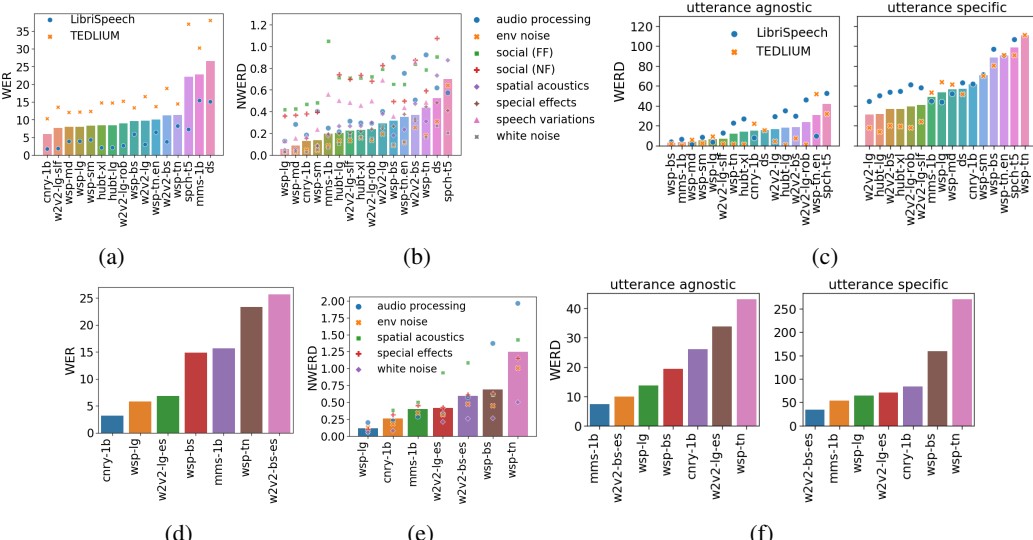

Figure 8: The accuracy and robustness of English (top) and Spanish (bottom) ASR models on clean and perturbed data. Accuracy is measured by WER of the models on clean speech (a & d). Robustness is measured by the NWERD on non-adversarially perturbed speech (b & e) and WERD on adversarially perturbed speech (c & f). The markers indicate the dataset in (a & c), and the perturbation category in (b & e). The x axes are in ascending order of the values on the y axes.

| lang | dataset | category model (abrv) | clean WER | accent | audio proc | noise (env) | noise (white) | sFX NWERD | social (FF) | social (NF) | spatial | synth speech | AVG | Adv (UA) WERD | Adv (US) | AVG |
|---|---|---|---|---|---|---|---|---|---|---|---|---|---|---|---|---|
| du | MLS-DU | cnry-1b | 4.0 | - | 15.5 | 5.3 | 6.0 | 12.4 | - | - | 12.2 | - | 10.3 | - | - | - |
| | | wsp-lg | 7.3 | - | 14.3 | 2.6 | 6.5 | 4.1 | - | - | 5.3 | - | 6.6 | - | - | - |
| | | mms-1b | 14.6 | - | 21.6 | 8.1 | 12.0 | 14.3 | - | - | 18.9 | - | 15.0 | - | - | - |
| | | wsp-bs | 20.4 | - | 93.8 | 14.0 | 26.3 | 34.5 | - | - | 24.7 | - | 38.6 | - | - | - |
| | LibriSpeech | prkt-rnnt-1.1b | 1.6 | - | 5.1 | 2.6 | 1.8 | 3.7 | - | - | 8.0 | 3.0 | 4.0 | - | - | - |
| | | cnry-1b | 1.7 | - | 12.3 | 3.0 | 2.7 | 12.7 | - | - | 9.3 | 4.2 | 7.4 | 7.7 | 61.7 | 34.7 |
| | | prkt-rnnt-0.6b | 1.8 | - | 11.2 | 3.4 | 3.0 | 6.0 | - | - | 9.4 | 3.8 | 6.1 | - | - | - |
| | | w2v2-lg-slf | 1.8 | - | 16.9 | 9.2 | 17.1 | 12.4 | - | - | 14.4 | 5.5 | 12.6 | 12.5 | 58.0 | 35.2 |
| | | prkt-ctc-0.6b | 2.0 | - | 10.9 | 3.5 | 3.4 | 7.3 | - | - | 10.4 | 4.7 | 6.7 | - | - | - |
| | | prkt-ctc-1.1b | 2.0 | - | 6.2 | 3.0 | 3.3 | 6.4 | - | - | 9.1 | 4.4 | 5.4 | - | - | - |
| | | hubt-xl | 2.1 | - | 18.2 | 11.1 | 16.3 | 13.3 | - | - | 14.7 | 5.5 | 13.2 | 26.7 | 54.4 | 40.5 |
| | | hubt-lg | 2.1 | - | 11.6 | 9.0 | 9.1 | 12.3 | - | - | 15.0 | 6.5 | 10.6 | 34.9 | 50.0 | 42.5 |
| | | sb-cnfmr | 2.6 | - | 9.4 | 11.0 | 15.2 | 11.3 | - | - | 19.0 | 7.4 | 12.2 | - | - | - |
| | | w2v2-lg-rob | 2.6 | - | 18.1 | 10.2 | 15.1 | 15.4 | - | - | 16.9 | 5.9 | 13.6 | 45.8 | 61.0 | 53.4 |
| | | sb-cnfmr-rnnt | 2.7 | - | 15.7 | 9.8 | 17.2 | 9.6 | - | - | 20.4 | 8.4 | 13.5 | - | - | - |
| | | w2v2-lg | 3.0 | - | 24.7 | 14.4 | 28.2 | 15.8 | - | - | 21.8 | 8.6 | 18.9 | 29.0 | 44.3 | 36.6 |
| | | w2v2-bs | 3.7 | - | 30.6 | 18.9 | 36.4 | 20.3 | - | - | 26.9 | 9.6 | 23.8 | 29.5 | 53.6 | 41.5 |
| | | wsp-md | 3.9 | - | 19.1 | 2.4 | 3.3 | 2.1 | - | - | 3.8 | 5.3 | 6.0 | 1.7 | 51.9 | 26.8 |
| | | wsp-lg | 3.9 | - | 6.9 | 2.0 | 3.5 | 1.6 | - | - | 3.0 | 4.0 | 3.5 | 3.4 | 43.6 | 23.5 |
| | | wsp-sm.en | 4.0 | - | 34.9 | 6.9 | - | 6.0 | - | - | 3.9 | 6.2 | 11.6 | - | - | - |
| | | wsp-md.en | 4.1 | - | 16.0 | 4.9 | - | 2.8 | - | - | 0.9 | 4.8 | 5.9 | - | - | - |
| | | wsp-sm | 4.3 | - | 27.9 | 4.0 | 5.4 | 4.5 | - | - | 6.5 | 7.0 | 9.2 | 8.3 | 71.1 | 39.7 |
| | | wsp-bs.en | 5.1 | - | 48.0 | 12.4 | - | 17.9 | - | - | 11.2 | 8.1 | 19.5 | - | - | - |
| | | wsp-bs | 5.9 | - | 63.6 | 6.6 | 11.0 | 14.8 | - | - | 14.5 | 8.5 | 19.8 | 3.6 | 96.7 | 50.2 |
| | | wsp-tn.en | 6.4 | - | 57.2 | 8.6 | 14.0 | 25.8 | - | - | 17.1 | 9.2 | 22.0 | 9.4 | 90.2 | 49.8 |
| | | spch-t5 | 7.2 | - | 32.6 | 49.8 | 26.2 | 27.3 | - | - | 67.9 | 64.3 | 44.7 | 52.4 | 106.4 | 79.4 |
| | | sbcrdnn | 7.2 | - | - | 11.5 | 33.9 | - | - | - | - | - | 22.7 | - | - | - |
| | | wsp-tn | 8.2 | - | 68.4 | 13.8 | 17.4 | 26.1 | - | - | 21.8 | 8.6 | 26.2 | 22.5 | - | 22.5 |
| | | ds | 15.1 | - | 37.9 | 26.2 | 28.9 | 32.0 | - | - | 46.2 | 18.0 | 31.5 | - | 62.9 | 62.9 |
| | | mms-1b | 15.4 | - | 16.3 | 5.5 | 7.0 | 11.1 | - | - | 11.9 | 8.9 | 10.1 | 6.2 | 44.6 | 25.4 |
| | TEDLIUM | cnry-1b | 10.2 | - | 15.3 | 3.1 | 2.9 | 10.6 | - | - | 11.7 | 1.5 | 7.5 | 21.9 | - | 21.9 |
| | | wsp-md | 12.0 | - | 22.9 | 1.7 | 3.5 | 3.1 | - | - | 4.3 | 0.8 | 6.1 | 5.7 | 61.3 | 33.5 |
| | | wsp-lg | 12.1 | - | 12.5 | 2.5 | 2.6 | 2.3 | - | - | 4.0 | 0.4 | 4.0 | 9.0 | 63.6 | 36.3 |
| | | wsp-sm | 12.3 | - | 32.0 | 2.3 | 4.9 | 6.1 | - | - | 5.6 | 1.6 | 8.8 | 2.2 | 69.8 | 36.0 |
| | | wsp-bs | 13.3 | - | 67.6 | 5.8 | 8.8 | 19.2 | - | - | 8.7 | 4.1 | 19.0 | 1.7 | 80.3 | 41.0 |
| | | w2v2-lg-slf | 13.5 | - | 27.6 | 9.5 | 19.3 | 17.2 | - | - | 18.8 | 9.9 | 17.0 | 1.5 | 24.0 | 12.8 |
| | | wsp-tn.en | 13.7 | - | 50.9 | 7.2 | 10.9 | 29.3 | - | - | 12.9 | 2.8 | 19.0 | 51.7 | 90.7 | 71.2 |
| | | wsp-tn | 14.4 | - | 59.7 | 11.5 | 15.8 | 29.9 | - | - | 16.2 | 4.3 | 22.9 | 1.3 | 110.9 | 56.1 |
| | | hubt-lg | 14.7 | - | 19.9 | 9.2 | 12.1 | 16.2 | - | - | 17.5 | 9.1 | 14.0 | 0.7 | 13.9 | 7.3 |
| | | hubt-xl | 14.7 | - | 24.6 | 11.1 | 17.3 | 16.9 | - | - | 18.2 | 8.2 | 16.0 | 1.2 | 19.1 | 10.2 |
| | | w2v2-lg-rob | 15.2 | - | 24.4 | 8.7 | 15.7 | 18.7 | - | - | 19.4 | 9.2 | 16.0 | 1.0 | 17.9 | 9.4 |
| | | w2v2-lg | 16.5 | - | 31.2 | 12.9 | 24.7 | 18.8 | - | - | 25.5 | 9.2 | 20.4 | 4.2 | 17.9 | 11.0 |
| | | w2v2-bs | 18.8 | - | 37.0 | 16.5 | 30.0 | 23.3 | - | - | 32.6 | 9.5 | 24.8 | 7.1 | 19.9 | 13.5 |
| | | mms-1b | 30.2 | - | 19.0 | 6.2 | 7.3 | 12.1 | - | - | 11.5 | 0.0 | 9.4 | 0.0 | 53.1 | 26.6 |
| | | spch-t5 | 37.0 | - | 48.8 | 31.8 | 15.2 | 27.5 | - | - | 48.1 | 21.7 | 32.2 | 31.8 | 90.6 | 61.2 |
| | | ds | 38.0 | - | 40.1 | 16.1 | 24.3 | 30.0 | - | - | 48.7 | 7.3 | 27.8 | 15.3 | 51.5 | 33.4 |
| | | cnry-1b | - | - | - | - | - | - | 31.7 | 13.9 | - | - | 22.8 | - | - | - |

| lang | dataset | model | c1 | c2 | c3 | c4 | c5 | c6 | c7 | c8 | c9 | c10 | c11 | c12 | c13 | c14 |
|---|---|---|---|---|---|---|---|---|---|---|---|---|---|---|---|---|
| | | hubt-lg | - | - | - | - | - | - | 51.0 | 27.7 | - | - | 39.4 | - | - | - |
| | | hubt-xl | - | - | - | - | - | - | 51.1 | 27.6 | - | - | 39.4 | - | - | - |
| | | mms-1b | - | - | - | - | - | - | 71.2 | 55.2 | - | - | 63.2 | - | - | - |
| | | prkt-ctc-0.6b | - | - | - | - | - | - | 34.5 | 14.5 | - | - | 24.5 | - | - | - |
| | | prkt-ctc-1.1b | - | - | - | - | - | - | 31.6 | 13.6 | - | - | 22.6 | - | - | - |
| | | prkt-rnnt-0.6b | - | - | - | - | - | - | 37.0 | 16.7 | - | - | 26.9 | - | - | - |
| | | prkt-rnnt-1.1b | - | - | - | - | - | - | 35.9 | 17.0 | - | - | 26.4 | - | - | - |
| | | sb-cnfmr | - | - | - | - | - | - | 63.0 | 35.4 | - | - | 49.2 | - | - | - |
| | | sb-cnfmr-rnnt | - | - | - | - | - | - | 58.6 | 31.6 | - | - | 45.1 | - | - | - |
| | | sbcrdnn | - | - | - | - | - | - | 91.2 | - | - | - | 91.2 | - | - | - |
| | | spch-t5 | - | - | - | - | - | - | 304.3 | 221.9 | - | - | 263.1 | - | - | - |
| | | w2v2-bs | - | - | - | - | - | - | 66.0 | 32.9 | - | - | 49.4 | - | - | - |
| | | w2v2-lg | - | - | - | - | - | - | 59.6 | 30.8 | - | - | 45.2 | - | - | - |
| | | w2v2-lg-rob | - | - | - | - | - | - | 51.1 | 24.7 | - | - | 37.9 | - | - | - |
| | | w2v2-lg-slf | - | - | - | - | - | - | 50.6 | 26.2 | - | - | 38.4 | - | - | - |
| | | wsp-bs | - | - | - | - | - | - | 39.5 | 19.1 | - | - | 29.3 | - | - | - |
| | | wsp-lg | - | - | - | - | - | - | 29.3 | 15.9 | - | - | 22.6 | - | - | - |
| | | wsp-md | - | - | - | - | - | - | 29.2 | 15.8 | - | - | 22.5 | - | - | - |
| | | wsp-sm | - | - | - | - | - | - | 31.4 | 17.0 | - | - | 24.2 | - | - | - |
| | | wsp-tn | - | - | - | - | - | - | 48.9 | 21.2 | - | - | 35.0 | - | - | - |
| | | wsp-tn.en | - | - | - | - | - | - | 41.7 | 19.5 | - | - | 30.6 | - | - | - |
| | | cnry-1b | - | - | - | - | - | - | 55.6 | 28.8 | - | - | 42.2 | - | - | - |
| | | ds | - | - | - | - | - | - | 90.6 | 84.0 | - | - | 87.3 | - | - | - |
| | | hubt-lg | - | - | - | - | - | - | 82.1 | 55.2 | - | - | 68.7 | - | - | - |
| | | hubt-xl | - | - | - | - | - | - | 81.1 | 54.6 | - | - | 67.8 | - | - | - |
| | | mms-1b | - | - | - | - | - | - | 126.7 | 98.8 | - | - | 112.7 | - | - | - |
| | | prkt-ctc-0.6b | - | - | - | - | - | - | 56.6 | 27.4 | - | - | 42.0 | - | - | - |
| | | prkt-ctc-1.1b | - | - | - | - | - | - | 53.4 | 26.7 | - | - | 40.0 | - | - | - |
| | | prkt-rnnt-0.6b | - | - | - | - | - | - | 60.5 | 28.3 | - | - | 44.4 | - | - | - |
| | | prkt-rnnt-1.1b | - | - | - | - | - | - | 55.9 | 27.3 | - | - | 41.6 | - | - | - |
| | | sb-cnfmr | - | - | - | - | - | - | 92.5 | 74.2 | - | - | 83.3 | - | - | - |
| | | sb-cnfmr-rnnt | - | - | - | - | - | - | 84.3 | 60.5 | - | - | 72.4 | - | - | - |
| | chime | sbcrdnn | - | - | - | - | - | - | 92.9 | 87.3 | - | - | 90.1 | - | - | - |
| | | spch-t5 | - | - | - | - | - | - | 527.7 | 388.7 | - | - | 458.2 | - | - | - |
| | | w2v2-bs | - | - | - | - | - | - | 88.7 | 64.6 | - | - | 76.7 | - | - | - |
| | | w2v2-lg | - | - | - | - | - | - | 86.9 | 61.7 | - | - | 74.3 | - | - | - |
| | | w2v2-lg-rob | - | - | - | - | - | - | 83.4 | 52.3 | - | - | 67.8 | - | - | - |
| | | w2v2-lg-slf | - | - | - | - | - | - | 81.6 | 52.0 | - | - | 66.8 | - | - | - |
| | | wsp-bs | - | - | - | - | - | - | 83.4 | 35.5 | - | - | 59.4 | - | - | - |
| | | wsp-lg | - | - | - | - | - | - | 48.6 | 21.6 | - | - | 35.1 | - | - | - |
| | | wsp-md | - | - | - | - | - | - | 50.6 | 22.8 | - | - | 36.7 | - | - | - |
| | | wsp-sm | - | - | - | - | - | - | 59.2 | 24.9 | - | - | 42.1 | - | - | - |
| | | wsp-tn | - | - | - | - | - | - | 98.6 | 46.3 | - | - | 72.4 | - | - | - |
| | | wsp-tn.en | - | - | - | - | - | - | 80.4 | 35.2 | - | - | 57.8 | - | - | - |
| | | cnry-1b | - | 4.6 | - | - | - | - | - | - | - | - | 4.6 | - | - | - |
| | | ds | - | 93.4 | - | - | - | - | - | - | - | - | 93.4 | - | - | - |
| | | hubt-lg | - | 14.1 | - | - | - | - | - | - | - | - | 14.1 | - | - | - |
| | | hubt-xl | - | 12.9 | - | - | - | - | - | - | - | - | 12.9 | - | - | - |
| | | mms-1b | - | 18.1 | - | - | - | - | - | - | - | - | 18.1 | - | - | - |
| | | prkt-ctc-0.6b | - | 5.8 | - | - | - | - | - | - | - | - | 5.8 | - | - | - |
| | | prkt-ctc-1.1b | - | 5.6 | - | - | - | - | - | - | - | - | 5.6 | - | - | - |
| | | prkt-rnnt-0.6b | - | 5.1 | - | - | - | - | - | - | - | - | 5.1 | - | - | - |
| | | prkt-rnnt-1.1b | - | 3.8 | - | - | - | - | - | - | - | - | 3.8 | - | - | - |
| | | sb-cnfmr | - | 20.3 | - | - | - | - | - | - | - | - | 20.3 | - | - | - |
| | | sb-cnfmr-rnnt | - | 19.1 | - | - | - | - | - | - | - | - | 19.1 | - | - | - |
| | CV | sbcrdnn | - | 44.8 | - | - | - | - | - | - | - | - | 44.8 | - | - | - |
| | | spch-t5 | - | 89.7 | - | - | - | - | - | - | - | - | 89.7 | - | - | - |
| | | w2v2-bs | - | 25.4 | - | - | - | - | - | - | - | - | 25.4 | - | - | - |
| | | w2v2-lg | - | 20.0 | - | - | - | - | - | - | - | - | 20.0 | - | - | - |
| | | w2v2-lg-rob | - | 13.9 | - | - | - | - | - | - | - | - | 13.9 | - | - | - |
| | | w2v2-lg-slf | - | 13.5 | - | - | - | - | - | - | - | - | 13.5 | - | - | - |
| | | wsp-bs | - | 10.0 | - | - | - | - | - | - | - | - | 10.0 | - | - | - |
| | | wsp-lg | - | 3.7 | - | - | - | - | - | - | - | - | 3.7 | - | - | - |
| | | wsp-md | - | 4.1 | - | - | - | - | - | - | - | - | 4.1 | - | - | - |
| | | wsp-sm | - | 6.5 | - | - | - | - | - | - | - | - | 6.5 | - | - | - |
| | | wsp-tn | - | 13.9 | - | - | - | - | - | - | - | - | 13.9 | - | - | - |
| | | wsp-tn.en | - | 12.7 | - | - | - | - | - | - | - | - | 12.7 | - | - | - |
| es | MLS-ES | cnry-1b | 3.2 | - | 16.4 | 9.6 | 8.2 | 16.0 | - | - | 10.9 | - | 12.2 | 26.1 | 84.3 | 55.2 |
| | | wsp-lg | 5.8 | - | 14.1 | 7.4 | 5.9 | 4.0 | - | - | 2.0 | - | 6.7 | 13.7 | 65.0 | 39.4 |
| | | w2v2-lg-es | 6.8 | - | 21.2 | 14.7 | 22.1 | 19.0 | - | - | 26.3 | - | 20.6 | 33.9 | 71.0 | 52.4 |
| | | wsp-bs | 14.8 | - | 90.9 | 22.7 | 27.3 | 32.5 | - | - | 18.0 | - | 38.3 | 19.5 | 159.5 | 89.5 |
| | | mms-1b | 15.7 | - | 18.5 | 19.7 | 37.1 | 20.0 | - | - | 14.9 | - | 22.0 | 7.4 | 53.8 | 30.6 |
| | | wsp-tn | 23.3 | - | 124.3 | 45.0 | 52.3 | 57.9 | - | - | 41.8 | - | 64.3 | 43.1 | 269.9 | 156.5 |
| | | w2v2-bs-es | 25.7 | - | 32.4 | 20.4 | 24.2 | 24.1 | - | - | 32.4 | - | 26.7 | 10.0 | 33.8 | 21.9 |
| | | cnry-1b | - | 205.0 | - | - | - | - | - | - | - | - | 205.0 | - | - | - |
| | | mms-1b | - | 7.7 | - | - | - | - | - | - | - | - | 7.7 | - | - | - |
| | | w2v2-bs-es | - | 13.3 | - | - | - | - | - | - | - | - | 13.3 | - | - | - |
| | CV:es | w2v2-lg-es | - | 9.3 | - | - | - | - | - | - | - | - | 9.3 | - | - | - |
| | | wsp-bs | - | 18.2 | - | - | - | - | - | - | - | - | 18.2 | - | - | - |
| | | wsp-lg | - | 2.0 | - | - | - | - | - | - | - | - | 2.0 | - | - | - |
| | | wsp-tn | - | 30.6 | - | - | - | - | - | - | - | - | 30.6 | - | - | - |
| fr | MLS-FR | cnry-1b | 6.1 | - | 15.2 | 5.2 | 7.3 | 13.0 | - | - | 10.0 | - | 10.1 | - | - | - |
| | | wsp-lg | 7.7 | - | 15.5 | 3.5 | 8.3 | 5.6 | - | - | 5.7 | - | 7.7 | - | - | - |
| | | mms-1b | 23.6 | - | 21.0 | 6.9 | 12.5 | 12.9 | - | - | 15.6 | - | 13.8 | - | - | - |
| | | wsp-bs | 26.0 | - | 124.8 | 16.7 | 44.4 | 47.9 | - | - | 25.9 | - | 51.9 | - | - | - |

Table 7: Accuracy and robustness of ASR models on all datasets

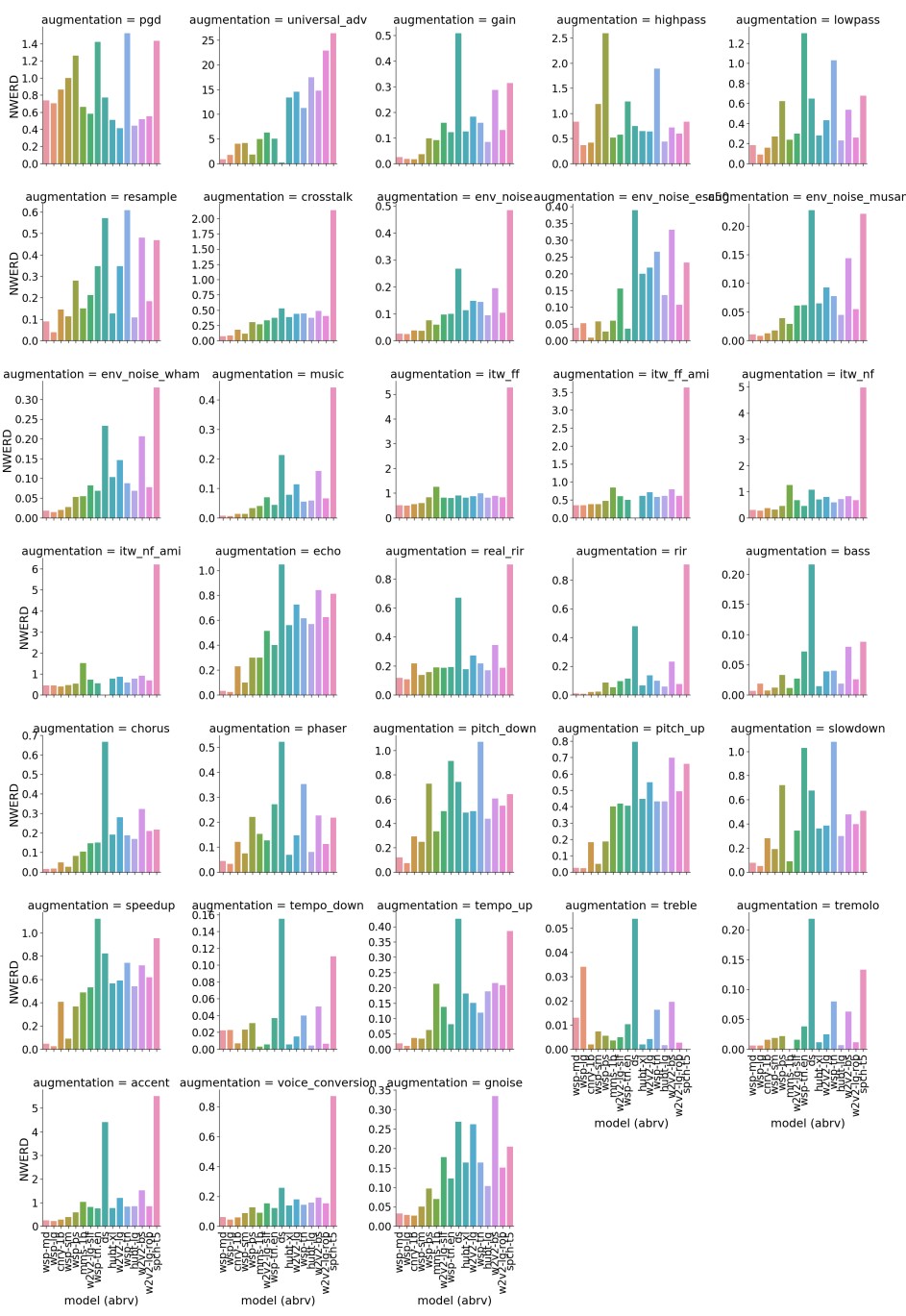

Figure 9: NWERD of English models on different augmentations, averaged over all severities.

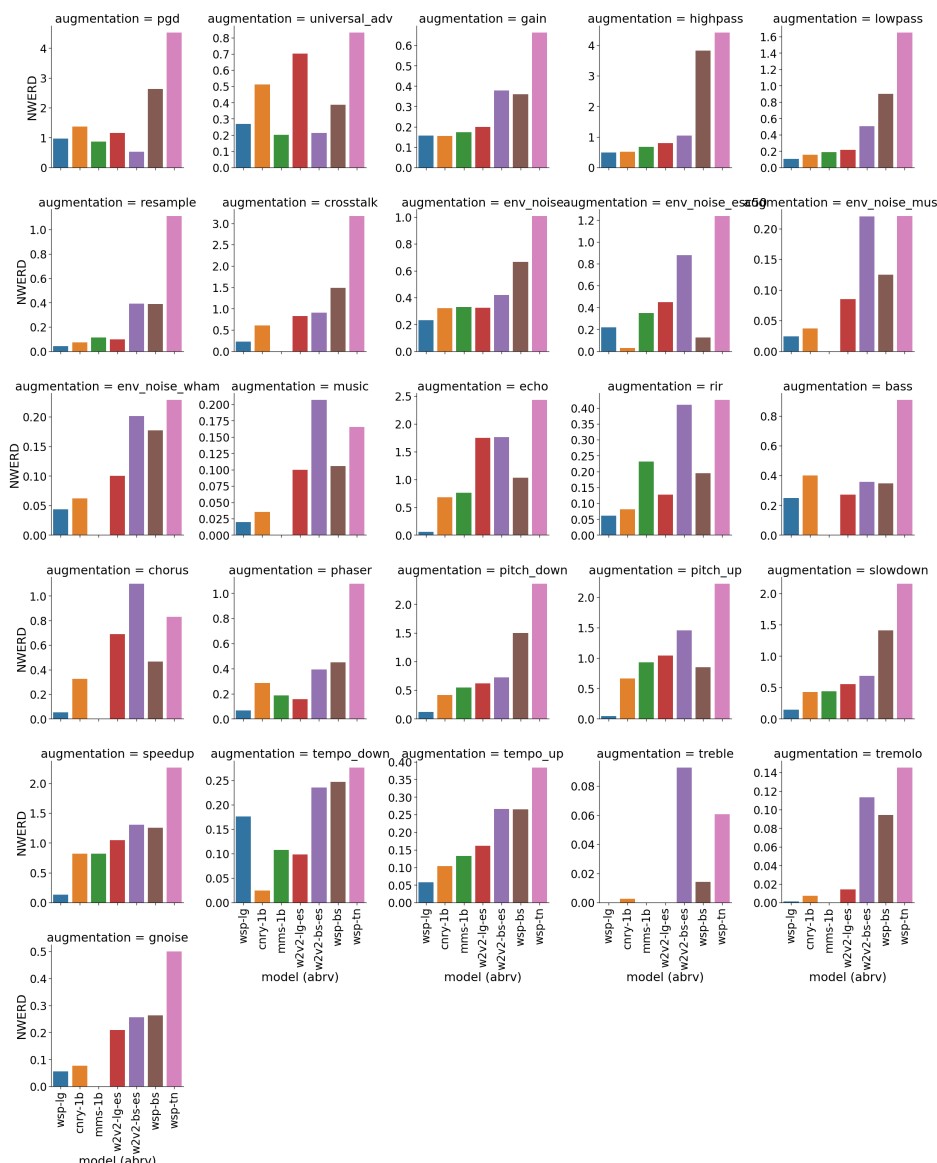

Figure 10: NWERD of Spanish models on different augmentations, averaged over all severities.

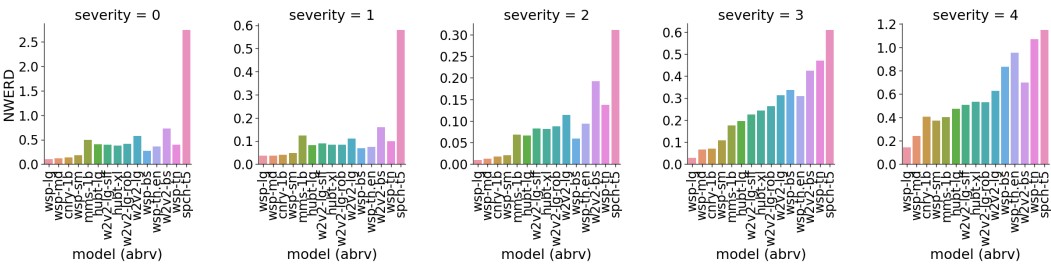

Figure 11: NWERD on English data as the severity of the augmentation is increased.

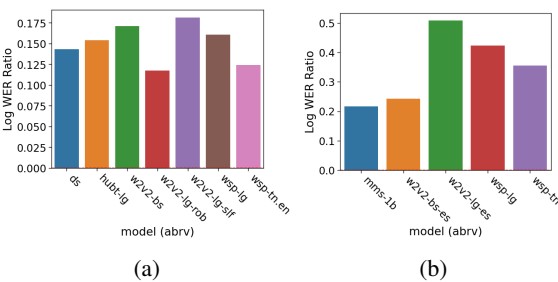

(a) (b)

Figure 12: Log WER Ratio between male and female speakers from Librispeech (English) (a) and Spanish Multilingual Librispeech (b).

