# OpenReview forum: "Speech Robust Bench: A Robustness Benchmark For Speech Recognition"
_ICLR.cc/2025/Conference — ICLR 2025 Poster_

### Official Review · Reviewer_Dmh5 · 2024-10-27

**Soundness:** 3
**Presentation:** 3
**Contribution:** 3
**Rating:** 6
**Confidence:** 4

**Summary:**

This paper proposes Speech Robust Bench, a benchmark framework for testing speech recognition models . It collects almost all speech variation techniques available in the field, with close connection with real-time application scenarios. Results on

**Strengths:**

1. The dataset description is very detailed and the related pipeline has been open-sourced, which is a decent contribution to the speech community.
2. The models covered in the field has been quite at the level of state-of-the-art.
3. The paper has answer to the critical question - "does the improvement come from additional variants from data or simply just MORE data?" in section 4.4, which is rare and good.

**Weaknesses:**

1. The reviewers think the presentation of the paper is good, but has room for improvement. Some of the extensive statistics and information (e.g. about attacks) can be enlisted or illustrated in the main text, rather than extensive text description in both main text and appendix; Some resulting figures (e.g. Figure 5) are hard to read; Also there are minor formatting issues.
2. One limitation of this work is lacking comparison or having initial attempts with some hybrid audio processing pipeline, especially based on its growing popularity on speech processing fields. In fact, applying noise and perturbation has been a long-time practice since Kaldi era.
3. It would be good if the authors can correspond the related earlier works onto the methods enlisted and highlight the differences and potential variants the authors covered.
4. The authors highlighted in the paper about multi-lingual scenario but in fact, only English and Spanish are covered, and the models are mostly mono-lingual. The authors may either remove such statement or want to conduct additional experiments using other languages and multi-lingual models for ASR (e.g. Google T5). This is not major problem though.

**Questions:**

The reviewer does not have explicit further questions apart from above commetns.

**Details Of Ethics Concerns:**

The reviewer thinks that this paper lacks claiming and concerning statements about the privacy and security of users' data.

The reviewer is not asking the authors to provide a solution but simply clarifying the responsibilities and open-sourcing, since this paper is about creating large-scale speech datasets.

---

> ### Author Response · Authors · 2024-11-25
>
> We thank the reviewer for their valuable feedback and are encouraged to see that they find our work valuable. Below we respond to each of the reviewer's questions and concerns. We hope that the reviewer will find our explanations satisfactory and will consider raising their score.
>
> **Some of the extensive statistics and information (e.g. about attacks) can be enlisted or illustrated in the main text,[...]; Some resulting figures (e.g. Figure 5) are hard to read**
>
> As per the reviewer’s suggestion, we have rewritten Section 3.1 to provide additional details about the various scenarios represented in SRB, including adversarial attacks. We have also increased the size of the figures that were hard to read. We hope that these changes address the reviewer’s concerns, however, if they do not we would be happy to make any additional updates that the reviewer suggests.
>
> **One limitation of this work is lacking comparison or having initial attempts with some hybrid audio processing pipelines[...]**
>
> We apologize, but it is not clear to us exactly what the reviewer is referring to. We would like to kindly as the reviewer to please elaborate on what they mean by a “hybrid audio processing pipeline” and if this involves augmentations during training or testing.
>
> **It would be good if the authors can correspond the related earlier works onto the methods enlisted and highlight the differences and potential variants the authors covered.**
>
> We apologize if we were not clear enough in communicating the novelty of SRB over existing benchmarks. We have modified section 2.3 to better convey the contributions of SRB, and we provide a summary below.
>
> The main contribution of SRB is that, unlike existing benchmarks, it enables comprehensive, fine-grained and standardized robustness evaluations of ASR models. It does this by addressing three key shotcomings of existing benchmarks:
> Many existing benchmarks are highly specialized and contain only one or few types of noises, and thus, individually, they do not evaluate robustness to the various challening scenarios that the models may encounter. For instance, CHiME and AMI datasets contain only speech from social environments, while WHAM!, ESC-50, MUSAN and MS-SNSD contain only environmental sounds and music, and likewise RobustSpeech only evaluates models on adversarial attacks. SRB, on the other hand, combines diverse types of noises, corruptions, and challenging speech recognition scenarios into a single benchmark that allows practitioners to evaluate the robustness of ASR models comprehensively. Furthermore, as mentioned in section 2.3, *SRB also contains certain perturbations and challenging speech recognition scenarios, namely special effects, accented speech and computer-generated speech, that are not included in existing benchmarks, and, thus, SRB provides is more comprehensive than the union of existing benchmarks.*
> A conseuqence of having several specialized benchmarks is that practitioners need to choose among several robustness benchmarks when evaluating their models. This is not only tedious, but also makes it difficult to compare the results of different studies if the benchmarks they use are different. Since SRB covers a wide-range of robustness related scenarios, practitioners would need to evaluate only on SRB and thus the results of various studies would be readily comparable.
> Benchmarks that try to mitigate the above two shortcomings are often too coarse to reveal the specific scenarios and corruptions the model(s) struggle against. For example, benchmarks like the Open ASR Leaderboard that evaluate on large datasets of real world speech, do not enable such fine-grained analysis of a model’s strengths and weaknesses because the sources of noise and distortion in each recording is not fully known. In contrast, the sources of noise, corruption and distortions present in each audio recordings in SRB is known, which allows users to pinpoint the weaknesses of the model during evaluation.
>
> **The authors highlighted in the paper about multi-lingual scenario but in fact, only English and Spanish are covered, and the models are mostly mono-lingual.**
>
>  We would to clarify that of the models we analyzed in section 4.5.1 only two (w2v-lg-es and w2v-bs-es) are monolingual spanish models, whereas the other 5 are multi-lingual models. Furthermore, Figure 6 explicitly compares robustness of multi-lingual models on English and Spanish speech. While it was not feasible for us to run extensive evaluations on a large number of languages, we have released the source code to allow practitioners to replicate SRB in their languages of choice. The released source code includes scripts for extracting accented speech in different languages from CommonVoice, and generating perturbed versions of any speech dataset on Huggingface. Thus SRB is easily extendable to other languages.
>
> We hope that the reviewer will find our explanations satisfactory and will consider raising their score.

---

> > ### Comment · Reviewer_Dmh5 · 2024-11-25
> >
> > Thanks to the author for the response. I think in terms of technical stuff, I will keep my score for now and look forward to further changes from the authors if the paper got accepted.
> >
> > Meanwhile, I wonder if there is any response from the author on ethical concern?

---

> > > ### Author Response · Authors · 2024-11-25
> > >
> > > Apologies for not including this in the original response.
> > >
> > > All the speech data included in this benchmark was obtained from publicly available datasets (LibriSpeech, Multilingual LibriSpeech, Common Voice, AMI, and CHiME). These datasets have been anonymized by their owners to ensure the privacy and security of the speakers involved. Since we did not record any speakers ourselves, we do not believe that we have any ethical obligations beyond ensuring that the aforementioned datasets are used in accordance with their respective licenses, which are mentioned in Table 6.
> > >
> > > We hope that this addresses the reviewer's concerns. Please feel free to follow up with further questions and comments if needed.

---

### Official Review · Reviewer_R2oE · 2024-10-29

**Soundness:** 2
**Presentation:** 2
**Contribution:** 2
**Rating:** 6
**Confidence:** 2

**Summary:**

This paper proposes SRB, a benchmark to evaluate the robustness of Automatic Speech Recognition (ASR) models. Extensive evaluations are conducted on several recent and popular ASR DNNs to analyze their robustness to non-adversarial and adversarial input perturbations,  and various sub-groups, namely English speech and non-English (Spanish) speech, and male and female speakers. From the above analysis, several conclusions are drawn.

**Strengths:**

1. The paper conducts extensive and detailed analysis to evaluate the robustness of recent ASR models. The Takeaways help develop robust ASR models.

**Weaknesses:**

1. While I appreciate the efforts to evaluate the robustness of recent ASR models, the paper offers limited novelty. I suggest that the authors compare their benchmarks to existing robustness metrics or frameworks in the ASR field.
2. To evaluate the robustness of ASR models, the current framework adds noise to clean speech data to evaluate the performance. How do recent ASR models perform in real-world noisy data? In other words, how can we evaluate the robustness of ASR models to real-world noisy data?
3. The paper lacks a discussion on strategies for improving robustness. It would be great if the authors could provide some sights or suggestions to improve the robustness.
2. The experimental sections of the paper need reorganization. I can not see a clear logic for the experimental parts.

**Questions:**

Listed in the weaknesses part

---

> ### Author Response · Authors · 2024-11-25
>
> We thank the reviewer for reading our paper closely and providing detailed feedback that will undoubtedly improve the clarity and impact of our work. Below we respond to each of the reviewer's questions and concerns. We hope that the reviewer will find our explanations satisfactory and will consider raising their score.
>
> **[...] the paper offers limited novelty. [...]**
>
> We apologize if we were not clear enough in communicating the novelty of SRB over existing benchmarks. We have modified section 2.3 to better convey the contributions of SRB, and we provide a summary below.
>
> The main contribution of SRB is that, unlike existing benchmarks, it enables comprehensive, fine-grained and standardized robustness evaluations of ASR models. It does this by addressing three key shotcomings of existing benchmarks:
> 1. Many existing benchmarks are highly specialized and contain only one or few types of noises, and thus, individually, they do not evaluate robustness to the various challening scenarios that the models may encounter. For instance, CHiME and AMI datasets contain only speech from social environments, while WHAM!, ESC-50, MUSAN and MS-SNSD contain only environmental sounds and music, and likewise RobustSpeech only evaluates models on adversarial attacks. SRB, on the other hand, combines diverse types of noises, corruptions, and challenging speech recognition scenarios into a single benchmark that allows practitioners to evaluate the robustness of ASR models comprehensively. Furthermore, as mentioned in section 2.3, *SRB also contains certain perturbations and challenging speech recognition scenarios, namely special effects, accented speech and computer-generated speech, that are not included in existing benchmarks, and, thus, SRB provides is more comprehensive than the union of existing benchmarks.*
> 1. A consequence of having several specialized benchmarks is that practitioners need to choose among them when evaluating their models. This is not only tedious but also makes it difficult to compare the results of different studies if the benchmarks they use are different. Since SRB covers a wide range of robustness-related scenarios, practitioners would need to evaluate models only on SRB and thus the results of various studies would be readily comparable.
> 1. Benchmarks that try to mitigate the above two shortcomings are often too coarse to reveal the specific scenarios and corruptions the model(s) struggle against. For example, benchmarks like the Open ASR Leaderboard that evaluate models on large datasets of real-world speech, do not enable such fine-grained analysis of a model’s strengths and weaknesses because the sources of noise and distortion in each recording is not fully known. In contrast, the sources of noise, corruption, and distortions present in each audio recording in SRB is known, which allows users to pinpoint the weaknesses of the model during evaluation.
>
> **[...]how can we evaluate the robustness of ASR models to real-world noisy data?[...]**
> We would like to clarify that, as mentioned in Figure 1 and section 3.1, SRB includes several types of challenging real-world speech recordings, including accented speech, recordings made in social settings like meetings and dinners using near and far-field microphones. Moreover, the environmental noise and room impulse responses used in SRB have been obtained from diverse real environments and are representative of real-world scenarios. Therefore, we believe that SRB can effectively measure the robustness of ASR models in challenging real-world scenarios.
>
>
> **The paper lacks a discussion on strategies for improving robustness.[...]**
>
> We have presented some insights along these lines in the section titled “Correlates of Robustness”, where we present results that indicate that models with more parameters tend to be more robust to both adversarial and non-adversarial perturbations, however, models trained on more data are not necessarily more robust. We also observe that CTC and RNNT models are more robust to adversarial attacks than seq2seq models and RNN-transducers are more robust to non-adversarial perturbations. These observations can be considered by model designers to arrive at more robust models.

---

> ### Author Response · Authors · 2024-11-25
>
> **The experimental sections of the paper need reorganization.[...]**
>
> The reviewer’s point regarding the organization of the experimental sections is well taken, and we have reorganized it to have a clearer logic. Specifically, we now have the following organization:
>
> 4 Evaluation
>
> |-- 4.1 Models
>
> |-- 4.2 Robustness of ASR Models
>
> |  |-- 4.2.1 Robustness in Non-Adversarial Scenarios
>
> |  |-- 4.2.2 Robustness in Adversarial Scenarios
>
> |-- 4.3 Correlates of Robustness
>
> |-- 4.4 Disparity in Robustness Across Population Subgroups
>
> |  |-- 4.4.1 Disparity in Robustness Across Languages in Multi-Lingual Models
>
> |  |-- 4.4.2 Disparity in Robustness Across Genders
>
> Section 4.2.1 and 4.2.2 combine the discussion of results on English and Spanish models.
> If the reviewer has any other suggestions about the ordering of Section 4, or any other aspect of the paper, we would be happy to update the paper accordingly.
>
> We hope that our responses adequately address the questions and concerns raised by the reviewer and that they would consider raising their score of our paper. If any concerns remain, we hope that the reviewer will raise them and give us the opportunity to address them.

---

> > ### Comment · Reviewer_R2oE · 2024-11-26
> > **Feedback to authors's response**
> >
> > I would like to thank the authors for their detailed responses. I have raised my rating from 5 to 6. I would also strongly suggest that authors consider open-sourcing the code if the paper is accepted.

---

> > > ### Author Response · Authors · 2024-11-26
> > >
> > > We want to thank the reviewer for going through our responses and we are encouraged to see that they have decided to increase their score. We also want to reiterate that we will release the full source code if the paper is accepted. An anonymized link to the code is present in the current manuscript as well.
> > >
> > > Since the reviewer is currently recommending marginal acceptance, perhaps there are aspects of the paper that could be improved. We would greatly appreciate it if the reviewer could point these out to us and give us the opportunity to improve upon them.
> > >
> > > Thanks

---

### Official Review · Reviewer_UsXF · 2024-10-30

**Soundness:** 3
**Presentation:** 3
**Contribution:** 2
**Rating:** 5
**Confidence:** 4

**Summary:**

This paper proposes a new benchmark data and pipeline to measure the ASR robustness under different perturbations. The proposed work has 4 components: 1. clean datasets and noise sources; 2. bank of perturbations; 3.ASR transcription extractions; 4. metric computations. Compared to existing benchmark pipeline, the authors claimed that “the proposed pipeline combines several benchmark datasets containing known noise sources to allow researchers to comprehensively evaluate the robustness of their models using a single benchmark”. During experiments, the authors evaluated a couple state-of-the-art ASR systems on the proposed benchmark, and conduct a series of analysis on robustness.

**Strengths:**

* The experiments are comprehensive. It considers a couple state-of-the-art ASR systems, and it analyzes adversarial/non-adversarial perturbations.
* It analyzed the correlation between different models in terms of robustness.
* It analyzed the robustness of speakers from different sub-groups (accent, gender).

**Weaknesses:**

Although the analyzes are very comprehensive, the contributions/novelty are not clearly stated. According to the quoted claim above, it feels like the most significant contribution compared to prior work is having more noise sources? Please explain more about the contributions/novelty here. It would be helpful to have a table to compare between the proposed approaches with baselines on the diffs from the 4 modules.

**Questions:**

See Weaknesses section

---

> ### Author Response · Authors · 2024-11-24
>
> We apologize if we were not clear enough in communicating the novelty of SRB over existing benchmarks. We have modified section 2.3 to better convey the contributions of SRB, and we provide a summary below.
>
> 1. The main contribution of SRB is that, unlike existing benchmarks, it enables comprehensive, fine-grained and standardized robustness evaluations of ASR models. It does this by addressing three key shotcomings of existing benchmarks:
> Many existing benchmarks are highly specialized and contain only one or few types of noises, and thus, individually, they do not evaluate robustness to the various challening scenarios that the models may encounter. For instance, CHiME and AMI datasets contain only speech from social environments, while WHAM!, ESC-50, MUSAN and MS-SNSD contain only environmental sounds and music, and likewise RobustSpeech only evaluates models on adversarial attacks. SRB, on the other hand, combines diverse types of noises, corruptions, and challenging speech recognition scenarios into a single benchmark that allows practitioners to evaluate the robustness of ASR models comprehensively. Furthermore, as mentioned in section 2.3, SRB also contains certain perturbations and challenging speech recognition scenarios, namely special effects, accented speech and computer-generated speech, that are not included in existing benchmarks, and, thus, SRB provides is more comprehensive than the union of existing benchmarks.
>
> 1. A conseuqence of having several specialized benchmarks is that practitioners need to choose among several robustness benchmarks when evaluating their models. This is not only tedious, but also makes it difficult to compare the results of different studies if the benchmarks they use are different. Since SRB covers a wide-range of robustness related scenarios, practitioners would need to evaluate only on SRB and thus the results of various studies would be readily comparable.
>
> 1. Benchmarks that try to mitigate the above two shortcomings are often too coarse to reveal the specific scenarios and corruptions the model(s) struggle against. For example, benchmarks like the Open ASR Leaderboard that evaluate on large datasets of real world speech, do not enable such fine-grained analysis of a model’s strengths and weaknesses because the sources of noise and distortion in each recording is not fully known. In contrast, the sources of noise, corruption and distortions present in each audio recordings in SRB is known, which allows users to pinpoint the weaknesses of the model during evaluation.
>
> We hope that our response adequately addresses the concerns of the reviewer regarding the novelty and contribution of our work and that they would consider raising their score of our paper. If there are any outstanding concerns, we encourage the reviewer to reply to us and we would be more than happy to address them.

---

> > ### Comment · Reviewer_UsXF · 2024-11-25
> > **Thank you for the responses!**
> >
> > Thank you for explaining on the novelty of this work! Raised my ratings accordingly.

---

> > > ### Author Response · Authors · 2024-11-25
> > >
> > > We thank the reviewer for taking the time to review our response and we are encouraged to see that that they have increased their score. However, it seems that we still fall short in some respects since the recommendation still is a marginal reject.
> > >
> > > We would appreciate it if the reviewer could elaborate on the specific aspects in which they find our work to be lacking and allow us the opportunity to make improvements.
> > >
> > > Thanks

---

### Official Review · Reviewer_4pNh · 2024-11-01

**Soundness:** 3
**Presentation:** 2
**Contribution:** 3
**Rating:** 6
**Confidence:** 3

**Summary:**

The authors propose a comprehensive benchmark for automatic speech recognition (ASR) models. The benchmark aims to cover a large variety of real-world data-quality challenges for deployed ASR models.

To benchmark on clean speech, the authors use the Librispeech and TEDLIUM datasets.
To benchmark on inter-personal speech, the authors use the CHiME-6 dataset.
To benchmark on far-field speech, the authors use the AMI dataset.
To benchmark on accented English, the authors use the Mozilla CommonVoice dataset.
To benchmark on text-to-speech audio, the authors generate data using the model XTTS.
To benchmark against adversarial attacks, the authors use utterance-specific and utterance-agnostic attacks.
To benchmark against environmental effects, the authors add 1) white noise, 2) environmental noise, or 3) simulate echo and RIR.
To benchmark against digital augmentations, the authors modify the audio data using most effects available in SoX.

The authors use the word-error-rate (WER) as the metric for clean speech. The metric for adversarial attacks is WER degredation (WERD). For all others, the authors use difficulty-normalized WER or WERD. The normalization is done based on a weight, which is an approximated speech quality given by a neural network.

The authors apply their benchmark on popular ASR models and their variants, including Whisper, wav2vec 2.0, HuBERT, and Canary.
The following observations are made:
- Canary has the best performance on clean speech
- Whisper is robust against digital augmentations, echo and RIR, while Canary is not.
- Wav2vec 2.0 is robust against adversarial attacks.
- There is a positive correlation between average robustness and amount of model parameters

Furthermore, the authors did an analysis on Spanish datasets with models fine-tuned (single-or multi-lingually) on Spanish. They find that multi-lingual models are more robust on English than Spanish.

Finally, the authors analyzed performance differences between male and female speakers. They observe it is data dependent.

**Strengths:**

### originality

This work is combines multiple existing datasets to build a comprehensive analysis tool for ASR models. The authors place their work fairly into existing literature.

### quality

The writing is mostly error-free and clear to understand. The authors analyzed relevant, contemporary ASR models. Nice-to-have but not required models to include could've been WavLM and, perhaps, OWSM.

### clarity

The authors use a familiar section lay-out for their paper. I also appreciate the take-away summaries in Section 4.

### significance

I think this work can contribute to a better evaluation of ASR models. It is significant to the speech community to make it is easy for researchers to evaluate on data other than Librispeech test-clean and test-other.

**Weaknesses:**

### Benchmark protocol is unclear

Section 3 is missing important details for reproducing this work. I think practitioners cannot replicate the benchmark protocol by reading the paper as-is. It is critical for the usefulness of a benchmark that it can be independently reproduced. I have the following remarks:

1. I find the usage of the word 'Perturbations' unclear. I see how adversarial attacks, environmental effects, and digital augmentations can be classified as perturbations of the audio signal. However, I do not see how accented speech, computer generated speech, and inter-personal communication, can be seen as a perturbation. I would prefer the usage of "domain" over "perturbation". My suggestion would be to change the taxonomy (Figure 2) to include the domains clean/professional, social gathering, accented, text-to-speech, noise (with subgroups environment, digital augmentation), and adversarial.
2.  It follows from 1 that is unclear to me exactly which dataset is used in which setting of the benchmark. I made a best guess in the summary above. For example, is environmental noise, or adversarial attacks, used only on Librispeech and TEDLIUM? Moreover, which split of Librispeech should be used? Only test-clean, or also test-other? Another example, which version of Mozilla CommonVoice is used? Do you filter out any data from CommonVoice if it is not labeled as 'accented'? I think the paper would be greatly improved by having a subsubsection for each domain/perturbation in Section 3.2, which list all details needed to generate the (exact) test set(s).
3. I praise the authors for sharing an (omited) link to perturbed versions of the datasets. I think this benchmark should require to use these exact perturbed versions in the noise domain, otherwise comparisons are unfair. The paper currently does not touch upon this topic.
4. Section 3.2 does not include any information on the generation of adversarial perturbations. These details should be in the main text of the paper.
5. There is no information on which text is used to generate the text-to-speech audio. Moreover, it would be desirable for these generated audio files to be shared so everyone can test on the same data.
6. Section 3.1 does not mention any Spanish datasets. Moreover, it is missing datasets for the noise domain (WHAM!, MUSAN, etc)
7. It is unclear to me whether this benchmark includes Spanish data, or whether this is simply an extra analysis from the authors.
8. For reproducing the benchmark, it is required to have the speech perception weights of each utterance to calculate the NWER(D).

### Confusing details in analysis

Section 4.1 mentions the use of DeepSpeech (which does not have open weights?) and speech-t5, these are not used in Table 1, only Figure 7. Moreover, Section 4.4 mentions the use of 4 more models, it would be more clear to include these in Section 4.1.

It is unclear when WER or WERD should be used. From Table 1 I assume that you use NWERD for e.g., accent and text-to-speech data. I don't see how one can have an $X$ and $X_p$ audio sequence to calculate the WERD in these scenarios.

### Figures and tables are difficult to read

I find Figure 4, 5 and 7 too small to read. As the page limit is 10, there should be enough room to increase the size without exceeding the page limit.

### Editorial remarks

* I would advice the authors to use \citep instead of \cite. Currently, the reference style (other than in section 2.2) makes the text hard to parse.
* Figure 3's space exceeds the bottom margin, and indents the texts on page 7 unnecessarily.
* Table 1 and Table 2 have a different format. I would recommend the use of the booktabs package and no usage of vertical lines.
* Appendix A is empty.
* Table 4 and Table 6 exceeds the page margin.
* The citation of Canary (Elena Rastogueva) is incorrectly formatted, is missing data, and exceeds the page margin.

#### Typos
- ln 41: is comphe.. -> are comphre...
- ln 107:  them -> those
- ln 161:  "others" does not fit here (Radford et al is in both)
- ln 346:  than -> compared to
- ln 472 (and others): the term accuracy is specific to classification, 'quality' is more appropriate to describe ASR predictions.

**Questions:**

1. Can the authors clarify, for each domain, what dataset(s) are used, specify their statistic (min/max/avg length, male/female, total number of utterances), exactly how a model is evaluated (WER, WERD, NWERD with speech quality weights from...) , and how the data is perturbed, if applicable?
2. Can the authors clarify whether Spanish data is included in the benchmark, or whether this is a separate analysis?
3. Can the authors clarify how weights for the DeepSpeech were obtained?

**Details Of Ethics Concerns:**

see comment to AE

---

> ### Author Response · Authors · 2024-11-24
>
> We thank the reviewer for reading our paper closely and providing detailed feedback that will undoubtedly improve the clarity and impact of our work. We have updated the manuscript after fixing the typos and editorial issues, as well as including the additional details that were suggested by the reviewer. Below we respond to each of the reviewer's questions and concerns. We hope that the reviewer will find our explanations satisfactory and will consider raising their score.
>
> **I find the usage of the word 'Perturbations' unclear […]**
>
> The reviewer’s suggestion is well taken and have replaced “perturbations” with “scenarios” as a general term referring to the different types of noises, corruptions and variations in SRB. We have rewritten Section 3.1 to frame SRB as simulating various challenging speech recognition scenarios, some of which, such as accents and inter-personal communication, are simulated using real noisy data or TTS, and the others by perturbing clean speechrecordings. We hope that this improves clarity and addresses the reviewer’s concerns.
>
>
> **[...] unclear to me exactly which dataset is used in which setting of the benchmark [...]**
>
> We apologize for the lack of clarity, and we have updated Section 3.1 to include more details about the scenarios. We also provide some of the details requested by the reviewer below. Adversarial attacks, environmental effects, and digital augmentations are applied to the test-clean subset of LibriSpeech, test subset of TEDLIUM release 3, and test subset of Spanish MultiLingual Librispeech. Computer-generated speech is also synthesized for the English datasets (LibriSpeech test-clean and TEDLIUM test). For accented speech, we used speech from CommonVoice 17. We used only the recordings which had an accent annotation and DNSMOS score >= 3.4. For English, we removed speakers with US English accents. We have added these details to Section 3.2. We have described the methodology of each perturbation in much greater detail in Appendix B to enable readers to replicate our settings. We have also linked our code repository in the paper which can be used to replicate our results. We would be more than happy to include additional details if the reviewer finds the current information to be inadequate
>
>
> **[...] this benchmark should require to use these exact perturbed versions in the noise domain [...]**
>
> The reviewer’s point is well taken and we have added a statement in Section 3 to encourage practitioners to use the exact perturbed data that we have made public. We also reiterate that we are releasing the code used to generate the benchmark datasets and thus users can easily recreate the benchmark using same datasets, or even other datasets that we have not currently used.
>
>
> **Section 3.2 does not include any information on the generation of adversarial perturbations.**
>
> We have added the objective functions optimized by the utterance-specific and utterance-agnostic attack in 3.2. The utterance-agnostic attack is, in principle, very similar to the utterance-specific attack, except it optimizes the adversarial perturbation over several speech recordings, instead of a single one. The full algorithm is presented in Algorithm 1.
>
>
> **There is no information on which text is used to generate the text-to-speech audio.[...]**
>
> The text-to-speech audio is generated English utterances from LibriSpeech test-clean and TEDLIUM release 3 test data, and Spanish speed. We have updated section 3.1 in the manuscript to reflect this. We have included the generated audio in the perturbed datasets that we will release after the double-blind restrictions are lifted.
>
>
> **Section 3.1 does not mention any Spanish datasets [...] datasets for the noise domain**
>
> We apologize for omitting details about the Spanish dataset, we have now updated section 3.1 with more details about the datasets and the various domains/scenarios present in SRB. We believe that the updated text provides sufficient information, nevertheless, we encourage the reviewer to respond with further suggestions for improvement and we would be happy to implement them.
>
>
> **It is unclear to me whether this benchmark includes Spanish data [...]**
>
> We apologize for the lack of clarity on our part. We have included the Spanish data (as well as French and German) in the perturbed datasets that we will publicly release to enable robustness evaluations for non-English and multilingual models. Table 2 is a demonstration of this use case. However, the comparison of English and Spanish performance of multilingual models is intended to be additional analysis and not necessarily part of the robustness benchmark.

---

> ### Author Response · Authors · 2024-11-24
>
> **it is required to have the speech perception weights of each utterance to calculate the NWER(D).**
>
> The reviewer is correct in pointing that perceptual metric values are required to compute NWER(D), therefore to facilitate this we have included these values in the code repository linked in the paper as well as in Table 4. We have also included scripts in the code repository to allow users to compute these metrics for themselves.
>
> **Section 4.1 mentions the use of DeepSpeech (which does not have open weights?) and speech-t5, these are not used in Table 1, only Figure 7. Moreover, Section 4.4 mentions the use of 4 more models, it would be more clear to include these in Section 4.1.**
>
> We apologize for this confusion and thank the reviewer for bringing this to our attention. We have now included the additional models from section 4.4 in 4.1.
>
> We would also like to clarify that Table 1 presents results from only a subset of all the models we evaluated. These models were selected based on their performance and popularity. Deepspeech and Speech-T5 had relatively weak performance so we decided to omit their results from Table 1 in the interest of clarity and brevity. We have mentioned this in section 4.2 of the updated manuscript.The results for these models, however, are included in Table 5. We would also like to clarify that DeepSpeech does have open weights and they can be found at https://github.com/SeanNaren/deepspeech.pytorch.
>
>
> **It is unclear when WER or WERD should be used.**
>
> We mentioned in the first paragraph of 3.1 that WER should be used to measure utility (read: accuracy) while WERD/NWERD should be used to measure robustness (i.e. loss in accuracy under challenging domains/perturbations). We have added a note on usage which clarifies that we use WERD to measure robustness against adversarial attacks, and NWER to measure robustness against all other perturbations. This is because adversarial attacks are model-specific and thus DNSMOS/PESQ scores for adversarially perturbed audio will be different for each model, which will lead to a different normalization during NWERD computation and make comparisons difficult.
>
>
> **I don't see how one can have an X and Xp audio sequence to calculate the WERD in these scenarios.**
>
> We apologize for lack of clarity on our part, and we have updated section 3.2 to further clarify how NWERD can be computed for TTS, accented speech, etc. Computing NWERD for TTS speech is straightforward because TTS speech is generated based on transcripts from LibriSpeech test-clean, TEDLIUM release 3 test set and Multilingual LibriSpeech Spanish test, so we use the original recordings from LibriSpeech, TEDLIUM and MultiLingual Librispeech as the set of reference audios X. When computing NWERD for naturally noisy or challenging speech like accented speech and speech from social gatherings we use the clean recordings from LibriSpeech as the set of reference audios X. We have added these details to Section 3.2
>
> **Can the authors clarify, for each domain, what dataset(s) are used, specify their statistic (min/max/avg length, male/female, total number of utterances), exactly how a model is evaluated (WER, WERD, NWERD with speech quality weights from...) , and how the data is perturbed, if applicable?**
>
> We have mentioned the datasets used for each domain and how the data is perturbed in Section 3.2 and Appendix B. We have also added the statistics that the reviewer has suggested in Table 4 in the appendix..
> As mentioned in Section 3.2, we use WER for clean recordings, WERD and NWERD for recordings from noisy scenarios. We have also added a note to section 3.2 clarifying that we use NWERD for non-adversarial scenarios and WERD for adversarial attacks. This is because adversarial attacks are model-specific and thus DNSMOS/PESQ scores for adversarially perturbed audio will be different for each model, which will lead to a different normalization during NWERD computation and make comparisons difficult.
> **Can the authors clarify whether Spanish data is included in the benchmark, or whether this is a separate analysis**
> The Spanish data is included in the benchmark and we will include a public link to the dataset in the paper after double blind restrictions are lifted. However, the comparison of the performance of multi-lingual models on English and Spanish was additional analysis that we did to demonstrate a use case of our benchmark.
>
> **Can the authors clarify how weights for the DeepSpeech were obtained?**
>
> Weights for DeepSpeech were obtained from https://github.com/SeanNaren/deepspeech.pytorch/releases/download/V3.0/librispeech_pretrained_v3.ckpt .
>
> We hope that our responses adequately address the questions and concerns raised by the reviewer and that they would consider raising their score of our paper.

---

> > ### Comment · Reviewer_4pNh · 2024-11-26
> >
> > I want to thank the authors for their response and their work updating the paper in such a short time-span. I've read the new version and most of my concerns were addressed.
> >
> > I still have a question regarding Section 3.2. The paper states (line 277):
> >
> > > we use WER Degradation (WERD), computed as WER($X_s$) − WER(X), where X and $X_s$ are datasets containing clean speech and speech from scenario $s$, respectively. For scenarios (1)-(3.1) (see § 3.1), $X_s$ is an inherently noisy dataset, and X
> > will be LibriSpeech for English and Multi-Lingual LibriSpeech for Spanish. For scenarios (3.2)-(6), X is a clean dataset, and $X_s$ is a perturbed version of X.
> >
> > It is now clear to me that you subtract the average WER over a _dataset_ from another average WER over _another dataset_, while I understood WER degredation to be in the context of an utterance, i.e., you add noise to an utterance and measure the difference in WER for that specfic utterance.
> >
> > Can the authors comment on their chosen approach of subtracting at a dataset level versus at an utterance level? For scenario 2 (social) and 3 (speech variation), I do not see why subtracting the Librispeech WER is required. For Scenario 4, 5, and 6, why is it better to subtract at the dataset level instead of doing the comparison at an utterance level like I described above?

---

> > > ### Comment · Reviewer_4pNh · 2024-11-26
> > >
> > > After thinking about it a bit more, I understand that for scenario 4, 5, and 6 it doesn't matter whether you average before or after the subtraction. For scenario 2 and 3, I think it's equally valuable to compare the numbers directly, although there is a case to be made to measure the relative difference w.r.t. librispeech performance.

---

> ### Author Response · Authors · 2024-11-26
>
> We thank the reviewer for taking the time to consider our responses and are happy to learn that we were able to address most of their concerns.
>
> With regards to the choice of subtracting LibriSpeech WER. We followed the advice of (Hendrycks & Dietterich, 2019) and measured robustness as the *relative degradation in accuracy/quality*, rather than the accuracy itself of the model under challenging scenarios. The reason behind this recommendation is that in many real-world cases, the stability of the model's accuracy under various scenarios may also be important, in addition to its best-/average-case accuracy. For example, consider two models such that they achieve 5% and 10% WER, respectively, on clean utterances, and, under a challenging scenario, both models achieve 11% WER. If only WER is used as a metric then both the models will seem equally robust. However, clearly, the first model's accuracy degraded by 6 times as much as the second model, which indicates that the second model is *much* more robust under the challenging scenario. Arguably, a user may prefer the second model despite slightly higher best-case WER because its WER does not vary significantly when the input becomes noisy and, thus, the user can better estimate the error and account for it in the downstream system.
>
> We hope that this explanation adequately addresses the reviewer's concerns about using WERD as a robustness metric, and that they would consider raising their score of our paper. If any concern or questions remain, please feel free to raise them in the follow up and we will be happy to provide further explanations.
>
> Dan Hendrycks and Thomas Dietterich. Benchmarking neural network robustness to common corruptions and perturbations. arXiv preprint arXiv:1903.12261, 2019.

---

### Author Response · Authors · 2024-11-25

We thank all the reviewers for taking the time to review our paper and providing valuable feedback that will surely help us improve the quality and impact of our work. We have provided explanations for each question and concern raised by each reviewer in "Official Comments" below the reviews. In our responses, we have quoted the **reviewer's comment in bold** and followed it with our explanation. We have also updated the manuscript based on the reviewers' suggestions. The salient changes are as follows:
1. Rewrote section 2.3 to enhance the clarity of presentation and emphasize the novelty and contributions of our benchmark over the existing robustness benchmarks for ASR models.
1. Included additional details in Section 3.2 for each scenario and perturbation used in SRB.
1. Editorial fixes such as typos, increasing figure sizes, and table formatting changes.
1. Reorganized Section 4 as follows:

4 Evaluation

|-- 4.1 Models

|-- 4.2 Robustness of ASR Models

| |-- 4.2.1 Robustness in Non-Adversarial Scenarios

| |-- 4.2.2 Robustness in Adversarial Scenarios

|-- 4.3 Correlates of Robustness

|-- 4.4 Disparity in Robustness Across Population Subgroups

| |-- 4.4.1 Disparity in Robustness Across Languages in Multi-Lingual Models

| |-- 4.4.2 Disparity in Robustness Across Genders

Section 4.2.1 and 4.2.2 combine the discussion of results on English and Spanish models.

We hope that our responses adequately address the questions and concerns raised by the reviewers and that they would consider raising their score on our paper. If any concerns remain, we hope that the reviewers will bring them to our attention and allow us to address them.

---

### Meta-Review · Area_Chair_UmfC · 2024-12-16

**Metareview:**

This work introduces a new benchmark dataset and pipeline designed to evaluate ASR robustness under various scenarios. The proposed framework consists of four key components: (1) clean datasets and noise sources, (2) a bank of perturbations, (3) ASR transcription extraction, and (4) metric computation. The benchmark aims to cover a large variety of real-world data-quality challenges for deployed ASR models.

The key authors' claim is that "the proposed pipeline integrates multiple benchmark datasets with known noise sources, enabling researchers to comprehensively assess the robustness of their models using a single unified benchmark."

Key strengths of the work are: (i) The experiments are thorough, evaluating several state-of-the-art ASR systems and examining both adversarial and non-adversarial perturbations,  (ii) The study explores the correlation between different models in terms of robustness and investigates the robustness of speakers from various sub-groups, including accent and gender, and (iii)  the authors conducts extensive and detailed analysis to evaluate the robustness of recent ASR models.

Key weakness is that multilingual scenario is not fully covered.

The proposed benchmark is different enough from what already available, and it gives researchers to comprehensively assess the robustness of their models using a single unified benchmark. The paper is well written.

**Additional Comments On Reviewer Discussion:**

Four reviewers assessed the work. The key concerns were about additional details about the experimental setup and clarification about novelty. The authors addressed those concerns effectively.

 Two reviewers flagged the work for ethics review. Specifically:

(1)  4pNh commented that "While skimming the Appendix, I noted that the Authors specify the use of particular compute cluster. The name of the cluster includes a city name. This identifies the Author's rough geographical area." However, that does not reveals the authors' identity and/or affiliation.
(2) Dmh5 stated that "this paper lacks claiming and concerning statements about the privacy and security of users' data."The following authors' answer seems to address the concern: "All the speech data included in this benchmark was obtained from publicly available datasets (LibriSpeech, Multilingual LibriSpeech, Common Voice, AMI, and CHiME). These datasets have been anonymized by their owners to ensure the privacy and security of the speakers involved. Since we did not record any speakers ourselves, we do not believe that we have any ethical obligations beyond ensuring that the aforementioned datasets are used in accordance with their respective licenses, which are mentioned in Table 6."

---

### Decision · Program_Chairs · 2025-01-22

Accept (Poster)